# Efficacy of Ivermectin, Chloroquine/Hydroxychloroquine, and Azithromycin in Managing COVID-19: A Systematic Review of Phase III Clinical Trials

**DOI:** 10.3390/biomedicines12102206

**Published:** 2024-09-27

**Authors:** Nathália Mariana Santos Sansone, Matheus Negri Boschiero, Fernando Augusto Lima Marson

**Affiliations:** 1Laboratory of Molecular Biology and Genetics, Laboratory of Clinical and Molecular Microbiology, LunGuardian Research Group—Epidemiology of Respiratory and Infectious Diseases, São Francisco University, Bragança Paulista 12916-900, SP, Brazil; nathalia.sansone@mail.usf.edu.br (N.M.S.S.); boschiero.matheus@gmail.com (M.N.B.); 2São Paulo Hospital, Federal University of São Paulo, São Paulo 04023-062, SP, Brazil

**Keywords:** antibiotics, azithromycin, clinical trial, chloroquine, coronavirus disease, hydroxychloroquine, ivermectin, severe acute respiratory syndrome coronavirus 2

## Abstract

**Background:** During the coronavirus disease (COVID)-19 pandemic several drugs were used to manage the patients mainly those with a severe phenotype. Potential drugs were used off-label and major concerns arose from their applicability to managing the health crisis highlighting the importance of clinical trials. In this context, we described the mechanisms of the three repurposed drugs [Ivermectin-antiparasitic drug, Chloroquine/Hydroxychloroquine-antimalarial drugs, and Azithromycin-antimicrobial drug]; and, based on this description, the study evaluated the clinical efficacy of those drugs published in clinical trials. The use of these drugs reflects the period of uncertainty that marked the beginning of the COVID-19 pandemic, which made them a possible treatment for COVID-19. **Methods:** In our review, we evaluated phase III randomized controlled clinical trials (RCTs) that analyzed the efficacy of these drugs published from the COVID-19 pandemic onset to 2023. We included eight RCTs published for Ivermectin, 11 RCTs for Chloroquine/Hydroxychloroquine, and three RCTs for Azithromycin. The research question (PICOT) accounted for P—hospitalized patients with confirmed or suspected COVID-19; I—use of oral or intravenous Ivermectin OR Chloroquine/Hydroxychloroquine OR Azithromycin; C—placebo or no placebo (standard of care); O—mortality OR hospitalization OR viral clearance OR need for mechanical ventilation OR clinical improvement; and T—phase III RCTs. **Results:** While studying these drugs’ respective mechanisms of action, the reasons for which they were thought to be useful became apparent and are as follows: Ivermectin binds to insulin-like growth factor and prevents nuclear transportation of severe acute respiratory syndrome coronavirus 2 (SARS-CoV-2), therefore preventing cell entrance, induces apoptosis, and osmotic cell death and disrupts viral replication. Chloroquine/Hydroxychloroquine blocks the movement of SARS-CoV-2 from early endosomes to lysosomes inside the cell, also, this drug blocks the binding between SARS-CoV-2 and Angiotensin-Converting Enzyme (ACE)-2 inhibiting the interaction between the virus spike proteins and the cell membrane and this drug can also inhibit SARS-CoV-2 viral replication causing, ultimately, the reduction in viral infection as well as the potential to progression for a higher severity phenotype culminating with a higher chance of death. Azithromycin exerts a down-regulating effect on the inflammatory cascade, attenuating the excessive production of cytokines and inducing phagocytic activity, and acts interfering with the viral replication cycle. Ivermectin, when compared to standard care or placebo, did not reduce the disease severity, need for mechanical ventilation, need for intensive care unit, or in-hospital mortality. Only one study demonstrated that Ivermectin may improve viral clearance compared to placebo. Individuals who received Chloroquine/Hydroxychloroquine did not present a lower incidence of death, improved clinical status, or higher chance of respiratory deterioration compared to those who received usual care or placebo. Also, some studies demonstrated that Chloroquine/Hydroxychloroquine resulted in worse outcomes and side-effects included severe ones. Adding Azithromycin to a standard of care did not result in clinical improvement in hospitalized COVID-19 participants. In brief, COVID-19 was one of the deadliest pandemics in modern human history. Due to the potential health catastrophe caused by SARS-CoV-2, a global effort was made to evaluate treatments for COVID-19 to attenuate its impact on the human species. Unfortunately, several countries prematurely justified the emergency use of drugs that showed only in vitro effects against SARS-CoV-2, with a dearth of evidence supporting efficacy in humans. In this context, we reviewed the mechanisms of several drugs proposed to treat COVID-19, including Ivermectin, Chloroquine/Hydroxychloroquine, and Azithromycin, as well as the phase III clinical trials that evaluated the efficacy of these drugs for treating patients with this respiratory disease. **Conclusions:** As the main finding, although Ivermectin, Chloroquine/Hydroxychloroquine, and Azithromycin might have mechanistic effects against SARS-CoV-2 infection, most phase III clinical trials observed no treatment benefit in patients with COVID-19, underscoring the need for robust phase III clinical trials.

## 1. Introduction

The coronavirus disease (COVID)-2019 was first reported in Wuhan, a Chinese province, and is caused by a novel species of coronavirus, namely severe acute respiratory syndrome coronavirus 2 (SARS-CoV-2) [1,2,3]. Due to the rapid spread of this novel virus, the World Health Organization (WHO) declared a pandemic status in March 2020. One of the utmost concerns about SARS-CoV-2 is its high potential to spread, with a global basic reproduction number (R_0_) varying from 3.42 to 4.08, which means one person with COVID-19 might transmit it to three to four other individuals [4]. Although the general case fatality rate (CFR) of COVID-19 is not high (nearly 1%), it is considerably higher in hospitalized patients, with a CFR of ~13% [5]. In this context, the combination of transmissibility and CFR might have made COVID-19 one of the deadliest pandemics in the world.

The COVID-19 pandemic has had a global impact affecting all spheres of society. In Public Health, it has led to an overload of health systems, loss of human lives, and impact on the mental health of the population [6,7,8]. In the economic scenario, there was a reduction in consumption and an increase in unemployment. In the social sphere, distancing caused the cancellation of social events. The pandemic has widened social inequalities, exposing the vulnerable population to contagion [6,7,8]. Due to the potential health catastrophe caused by SARS-CoV-2, a global effort was made to evaluate treatments for COVID-19 to attenuate its global impact, especially by the WHO, with the RECOVERY (Randomized Evaluation of COVID-19 Therapy) international trial [9]. Unfortunately, several countries, such as the United States of America (USA) and Brazil, started to use drugs that showed only in vitro effects against the SARS-CoV-2, such as Ivermectin, Chloroquine/Hydroxychloroquine (CQ/HCQ), and Azithromycin [6,10,11,12]. Due to the pandemic emergency, the use of well-known substances in order to treat a new disease was a valid option in the beginning, which is called repurposed drugs [13]. Many politicians and healthcare workers who have advocated the early use of these drugs used the argument of lack of time to pursue proper randomized clinical trials (RCTs). Hence, the emergency use of those drugs was justifiable [14]. However, most of the pre-clinical phase I and II studies findings are not confirmed in phase III clinical studies, mainly due to the fact the null hypothesis has a higher probability than the alternative hypothesis being true [14,15].

Until 2024, COVID-19 accounted for approximately 774 million confirmed cases and more than seven million confirmed deaths worldwide [16]. Almost four years after the COVID-19 pandemic, several robust and methodological RCTs have been published regarding the efficacy of Ivermectin [17,18,19,20,21,22,23,24], CQ/HCQ [19,25,26,27,28,29,30,31,32,33,34], and Azithromycin [35,36,37]. In Brazil, several physicians, politicians, the Ministry of Health, and the Federal Medicine Council insist on using these drugs to treat COVID-19, with the creation of an app by the Ministry of Health named “TrateCov”, which recommends the Ivermectin and the HCQ [6].

Thus, we aimed (i) to evaluate the physiological mechanisms of Ivermectin, CQ/HCQ, and Azithromycin, which made them a possible treatment for inpatients with COVID-19 (scientific rationale to repurpose these drugs); and (ii) to describe the results of phase III RCTs which evaluate the efficacy of these drugs to treat the patients with COVID-19 using a systematic review approach. Other medications, such as Remdesvir, were excluded because they received emergency approval from several regulatory agencies.

## 2. Materials and Methods

We searched the PubMed-Medline (Medical Literature Analysis and Retrievel System Online), Cochrane, and SciELO (Scientific Electronic Library Online) databases using the following descriptors: “Azithromycin”, “Antibiotics”, “Chloroquine”, “COVID-19”, “COVID-19 treatments”, “COVID-19 pandemic”, “Hydroxychloroquine”, “Ivermectin”, “Macrolides”, “SARS-CoV-2”, and “SARS-CoV-2 infection”.

We included phase III RCTs from the COVID-19 pandemic onset to 2023. We did not include pre-print studies. Also, the mechanisms of Ivermectin, CQ/HCQ, and Azithromycin, which made them a possible treatment for COVID-19, were obtained in the literature. In brief, our research question (PICOT) accounted for P—hospitalized patients with confirmed or suspected COVID-19; I—use of oral or intravenous Ivermectin OR CQ/HCQ OR Azithromycin; C—placebo or no placebo (standard of care); O—mortality OR hospitalization OR viral clearance OR need for mechanical ventilation OR clinical improvement; and T—phase III RCTs.

We excluded the following article types: (i) not related to COVID-19; (ii) not phase III clinical trial; (iii) did not use any of the studied drugs (CQ/HCQ, Ivermectin or Azithromycin) orally or intravenous; (iv) retracted articles; (v) was not published in English; (vi) did not comprise the date range of the study (COVID-19 pandemic onset to December 2023); (vii) did not evaluate inpatients with COVID-19; (viii) prophylaxis studies; (ix) if we did not have access to the article; (x) if the trial was not registered in online platforms (such as clinicaltrials.com); and (xi) did not present the outcome of interest. In addition, the complete selection of the studies is presented in the result section per drug included in the review.

The data search was performed on PubMed-Medline, Cochrane, and SciELO from the COVID-19 pandemic onset to December 2023. The following searches were performed:

(a) Search—Ivermectin: (Ivermectin) AND (“COVID-19” OR “COVID-19 treatments” OR “COVID-19 pandemic” OR “SARS-CoV-2” OR “SARS-CoV-2 infection”) AND (Therapy/Narrow[filter]) AND (randomized controlled trial[pt] OR controlled clinical trial[pt] OR clinical trials as topic[mesh:noexp] OR trial[ti] OR random*[tiab] OR placebo*[tiab]);

(b) Search—CQ/HCQ: (Chloroquine OR Hydroxychloroquine) AND (“COVID-19” OR “COVID-19 treatments” OR “COVID-19 pandemic” OR “SARS-CoV-2” OR “SARS-CoV-2 infection”) AND (randomized controlled trial[pt] OR controlled clinical trial[pt] OR clinical trials as topic[mesh:noexp] OR trial[ti] OR random*[tiab] OR placebo*[tiab]);

(c) Search—Azithromycin: (Azithromycin OR Antibiotics OR Macrolides) AND (“COVID-19” OR “COVID-19 treatments” OR “COVID-19 pandemic” OR “SARS-CoV-2” OR “SARS-CoV-2 infection”) AND (Therapy/Narrow[filter]) AND (randomized controlled trial[pt] OR controlled clinical trial[pt] OR clinical trials as topic[mesh:noexp] OR trial[ti] OR random*[tiab] OR placebo*[tiab]).

The bias risk assessment of the studies was made by using the Revised Tool for Assessing Risk of Bias in Randomized Trials (RoB 2), as recommended by Cochrane [38]. All the studies received the score “high risk”, “low risk” or “some concerns risk” for all of the five domains described in the Cochrane tool: (i) randomization; (ii) deviation from planned interventions; (iii) measurement of outcomes; (iv) selection of reported results; and (v) missing outcomes data. In addition, two independent authors (M.N.B. and F.A.L.M.) were involved in study selection and data extraction; also, in cases where there was disagreement between the authors, a third researcher (N.M.S.S.) was contacted to contribute to resolving the doubt.

This study was performed according to the Preferred Reporting Items for Systematic Reviews and Meta-Analyses statement (PRISMA) [39,40]. The figures from the study were created with BioRender.com.

## 3. Results

### 3.1. Ivermectin

Ivermectin (Figure 1), a worldwide anti-helminth drug [41], is a derivative from the 16-membered macrocyclic lactone, belonging to the Avermectin family [42,43], being first discovered in the Japan Kitasato Institute in 1967, with its first approved for use only 20 years later, in 1987, to treat onchocerciasis [43]. It also showed effects on nematodes, mites, and insects [43]. However, due to its in vitro antiviral effect against Ribonucleic Acid (RNA) and Deoxyribonucleic Acid (DNA) viruses, it was hypothesized to be effective in treating the SARS-CoV-2 infection [43,44]. Although Ivermectin has a safe profile, its excessive use was associated with several adverse effects, such as diarrhea, dizziness, abdominal pain, and vomiting, or even more severe adverse events, such as lethargy and dizziness, or even coma [45,46,47].

#### 3.1.1. Antiviral Mechanism of the Ivermectin

The antiviral mechanism of Ivermectin may be due to several different action sites, such as direct action on the SARS-CoV-2, on host sites essential for viral replication, on host targets important for inflammation, and even on other host important targets [48]. The antiviral mechanism is presented in Figure 1.

Ivermectin can disrupt the binding of essential proteins that allow cell entrance, such as Transmembrane Serine Protease 2 (TMPRSS2) and the Spike Protein in silico model [49]. Ivermectin was also described to (i) bind to the alpha subunit of the insulin-like growth factor superfamily and prevent the nuclear transportation of the SARS-CoV-2, and (ii) generate apoptosis and osmotic cell death by upregulating chloride channels since Ivermectin molecules behave as ionophores; however, no studies evaluated this effect on SARS-CoV-2 infection, only in leukemia cells [48,50,51,52]. In the same way, Ivermectin was able to bind to essential proteins for viral replication, such as nonstructural protein 1 (nsp-14) and Karyopherin-α1 (KPNA1), thus decreasing viral replication activity [11,48,53]. Ivermectin also plays a vital role in several pro-inflammatory and anti-inflammatory cytokines, as inhibition of Toll-Like Receptors (TLRs), especially the TLR-4, blockade the nuclear factor kappa-light-chain-enhancer of activated B cells (NF-kB) transcriptional pathway, decreased the expression of Tumor Necrosis Alpha (TNF-α) and Interleukin (IL)-6, enhance the expression of Interferon (IFN) related genes, as Interferon Induced Protein with Tetratricopeptide Repeats 1, Interferon Induced Protein with Tetratricopeptide Repeats 2, Interferon-Induced Protein 44, Interferon-Induced Protein 20, Interferon Regulatory Factor 9, and Oligoadenylate Synthase which might “protect” the host cell from the SARS-CoV-2 infection [48,54,55,56]. Curiously, the concentration of Ivermectin necessary to reach antiviral activity in the study of Caly et al. (2020) was 5 µM [11]; however, the maximum plasma concentration observed in vivo was 0.28 µM [57,58,59]. In that sense, Ivermectin is unlikely to have antiviral activity in vivo due to low concentration.

#### 3.1.2. Efficacy of Ivermectin to Treat Coronavirus Disease (COVID)-19 in Randomized Controlled Trials (RCTs)

The majority of studies found no significant benefit of Ivermectin in reducing mortality or severity in patients with COVID-19. In this context, in the systematic review, we obtained a total of 332 studies using the descriptors we described above. Of those studies, 94 were excluded for being duplicates. We also excluded 230 studies that did not meet the inclusion criteria, as described in Figure 2. We included a total of eight phase III RCTs that met the inclusion criteria [17,18,19,20,21,22,23,24]. In Table 1, we assessed all available phase III RCTs which evaluated Ivermectin as a treatment against COVID-19.

Lim et al. (2022) enrolled 490 participants with SARS-CoV-2 real-time-polymerase chain reaction (RT-PCR) positive test, older than 50 years, with at least one comorbidity who presented mild to moderate COVID-19 out of which 241 participants were assigned to receive 6- to 12-mg of Ivermectin plus standard of care and 249 participants were set to the standard of care group for five days [21]. This study’s standard of care consisted of symptomatic therapy and monitoring signs [21]. The primary outcome was progression to severe COVID-19, defined as a hypoxic stage requiring supplemental oxygen to maintain peripheral oxygen saturation (SpO_2_) >95% [21]. No difference was observed in the comparison between both groups (Ivermectin plus standard of care versus standard of care group alone) [21]. In this context, Ivermectin did not prevent the progression to severe COVID-19 in this study. However, 13.7% of the participants in the Ivermectin group reported any adverse event, whereas only 4.4% reported in the control group (standard of care group), which was the most common diarrhea [21].

Rezai et al. (2022) enrolled 891 no hospitalized and hospitalized adults with a positive SARS-CoV-2 RT-PCR, with moderate clinical symptoms [23]. Only 609 participants [311 individuals in the Ivermectin arm (0.4 mg/kg of body weight per day for three days) and 298 in the control arm] completed seven days of follow-up [23]. The primary outcome was time to resolution of symptoms, recovery including complete recovery (resolving main complaints on the sixth day) and relative recovery (remaining main complaints on the sixth day); disease progression (needing hospitalization), and negative SARS-CoV-2 RT-PCR result at five days [23]. In the study, complete recovery was higher in the Ivermectin group (37%) compared to the placebo group (28%), as well as the length of hospital stay was longer in the Ivermectin group (7.98 ± 4.4 days versus 7.16 ± 3.2 days) [23]. On the seventh day of treatment, fever, cough, and weakness were higher in the placebo group compared to the Ivermectin group [23]. Among all outpatients, 7% in the Ivermectin group and 5% in the placebo group needed to be hospitalized, and the result of SARS-CoV-2 RT-PCR on day five after treatment was negative for 26% of participants in the Ivermectin group versus 32% in the placebo group [23]. Ivermectin did not have a significant potential effect on clinical improvement, reduction in admissions in an intensive care unit, need for invasive ventilation, and number of deaths in hospitalized participants [23]. Also, no evidence was found to support the efficacy of Ivermectin on clinical recovery, decreased hospital length of stay, and increased negative RT-PCR assay for SARS-CoV-2 five days after treatment in outpatients [23].

A smaller study conducted by Baghbanian et al. (2023) evaluated a total of 60 intubated individuals with COVID-19, being that 31 subjects were treated with 6-mg of Ivermectin twice a day on the first day, followed by 3-mg twice a day from days two to five, whereas 29 subjects were part of the placebo group [24]. The primary outcome was in-hospital mortality [24]. The mortality rate was similar in both groups in days following hospitalization and days following intubation being that eventually, all of the participants in both groups died [24]. The authors concluded the use of Ivermectin in intubated individuals with COVID-19 was not associated with decreased mortality [24].

In a phase III RCT that took place in Mexico, Beltran Gonzalez et al. (2022) evaluated a total of 106 hospitalized individuals with confirmed or suspected COVID-19 pneumonia [19]. Out of the 106 individuals, 33 subjects received HCQ which will be addressed further in this paper, while 36 subjects received 12-mg (<80 kg) to 18-mg (>80 kg) of Ivermectin according to patients’ weight and 37 participants received a placebo [19]. It is worth mentioning that due to the RECOVERY trial [60], all the participants who needed oxygen support received dexamethasone 6-mg intravenous once per day for 10 days or until discharged during the last week of June 2020 [19]. The primary outcome was a composite of length of hospital stay, death, and respiratory deterioration [19]. No difference in hospital stay was observed in the Ivermectin group [6 days; interquartile range (IQR) = 4 to 11 days] compared to the placebo (5 days; IQR = 4 to 7 days), nor in respiratory deterioration or death [8/36 (22.2%) versus 9/37 (24.3%)] nor death alone [5/36 (13.8%) versus 6/37 (16.2%)] [19]. The authors concluded that there was no benefit in using Ivermectin to treat hospitalized individuals with confirmed or suspected COVID-19 [19].

A phase III clinical trial conducted in Iran by Shakhsi Niaee et al. (2021) evaluated 180 hospitalized individuals with mild COVID-19 confirmed by RT-PCR or with compatible chest images [18]. All of the participants were equally allocated in six arms: (i) HCQ 200-mg twice per day; (ii) placebo plus HCQ 200-mg twice per day; (iii) a single dose of Ivermectin 200 mcg/kg; (iv) three low interval dosages of Ivermectin (200, 200, and 200 mcg/kg); (v) a single dose of Ivermectin (400 mcg/kg); and (vi) three high interval doses of Ivermectin (400, 200, and 200 mcg/kg) [18]. The primary endpoint accounted for all-cause mortality or clinical recovery [18]. The authors observed that the individuals enrolled in arms iii (no death), iv (3/30, 10%), v (no death), and vi (1/30, 3.3%) were less likely to die compared to individuals from arms i (5/30, 16.7%) and ii (6/30, 20%) [18]. The authors concluded that Ivermectin may be beneficial in treating hospitalized individuals with confirmed or suspected COVID-19 [18]. However, it is important to highlight that the demographic baseline characteristics of the participants may not have been homogenous such as body mass index and SpO_2_ differed significantly between groups, which might have been due to impaired randomization [18]. Not only that, the study did not provide data regarding comorbidities such as diabetes mellitus, smoking status, or any characteristic that might play a significant role in COVID-19 mortality [61], which ultimately could have biased the results.

In another study from the Middle East, in Pakistan, Qadeer et al. (2022) evaluated 210 individuals with COVID-19 who were treated in a COVID-19 treatment center [22]. A total of 105 individuals received 12-mg of Ivermectin once daily for five days plus standard of care whereas 105 participants received only standard of care [22]. The primary outcome was the time of viral clearance measured by RT-PCR on days seven, 14, and 21 [22]. A total of 21 (20%) participants of those in the Ivermectin group had a negative RT-PCR on day seve, while in the placebo group, all of the 105 individuals still tested positive for COVID-19. On day 10, 70 (66.7%) participants in the Ivermectin group had negative RT-PCR for COVID-19 versus 21 (20%) individuals in the placebo group [22]. On day 14, all of the participants in the Ivermectin group tested negative versus 70 (66.7%) individuals in the placebo group [22]. The authors concluded that Ivermectin may improve viral clearance compared to placebo [22].

Heydari et al. (2022) conducted a phase III RCT with 107 hospitalized individuals with confirmed COVID-19 by RT-PCR or chest image [20]. The trial accounted for three arms, being 44 participants allocated to receive 200 mcg/kg (maximum of 12-mg) of Ivermectin, 17 participants received eight mg/kg of metronidazole four times a day for five days plus standard of treatment, and 44 participants received only standard treatment [20]. The primary outcome was (i) vital signs (body temperature, respiratory rate, heart rate, systolic blood pressure, diastolic blood pressure, and SpO_2_), (ii) biomedical parameters such as the levels of lymphocytes, neutrophils, platelets, and white blood cells, and (iii) length of hospital stay and death [20]. The mortality rate in Ivermectin was lower compared to the other groups (4.5% versus 15.8% versus 11.8%), but not statistically significant [20]. After five days, the mean difference in lymphocyte and neutrophil count was significantly different between groups [20]. The other characteristics were not significant [20]. Ivermectin did not improve patients’ recovery compared to standard of care alone, and it also did not improve hospital length of stay, and mortality [20].

In the same way, Okumuş et al. (2021) evaluated 66 hospitalized individuals with COVID-19 severe pneumonia [17]. A total of 30 participants received standard treatment plus 200 mcg/kg of Ivermectin for five days whereas 30 individuals received only standard treatment [17]. The primary outcome was the clinical response on the fifth day [extubating rates of mechanically ventilated individuals, respiratory rate <26 beats per minute, SpO_2_ level in room air >90%, ratio of partial pressure of oxygen in arterial blood (PaO_2_) to the fraction of inspiratory oxygen concentration (FiO_2_) (PaO_2_/FiO_2_) >300 in individuals receiving oxygen, presence of at least two of the 2-point reduction criteria in Sequential Organ Failure Assessment (SOFA)] and drug side effects [17]. At the end of the 5-day follow-up: 22/30 (73.3%) of the individuals improved compared to 16/30 (53.3%) in the control group [17]. Regarding mortality, 6 (20%) of the participants receiving Ivermectin died compared to 9 (30%) in the control group [17]. The differences in the primary outcome were not statistically significant [17]. In this context, although the authors conclude that Ivermectin may be an optional treatment for COVID-19, which might improve clinical response and mortality rate, their data suggest otherwise, since the difference between groups was not statistically significant [17].

A recent meta-analysis of RCTs of Ivermectin to treat individuals with COVID-19 performed by the Cochrane Library observed that Ivermectin did not decrease all-cause mortality in up to 28 days in in-patients and out-patients and did not prevent the need for mechanical ventilation [62]. However, most of the studies included in this meta-analysis had low or very low certainty of the evidence, making it challenging to analyze and draw conclusions about the real efficacy of Ivermectin during the COVID-19 pandemic.

Most studies, and even a recent meta-analysis, found no significant benefit of Ivermectin in reducing mortality or severity in patients with COVID-19 [17,19,20,21,24,62] even though Ivermectin appears to have mechanisms to prevent infection as an antiviral drug, such as preventing the viral replication and entrance into a host cell of the SARS-CoV-2 and also promoting osmosis [12,48,49,50,51]. Thus, it would be prudent not to embrace the alternative hypothesis (H1), even though most of these articles had low to very low certainty of the evidence, and stick to the null hypothesis (H0), that is, Ivermectin does not show a significant effect on the treatment of COVID-19.

Regarding the Risk of bias, most of the studies had low methodological quality, as shown in Table 2 most of the studies had a high risk of bias or at least some concerns (6/8). Only a few studies were classified as low risk of bias (2/8). This ultimately shows how the *paperdemic* [63] of COVID-19 might have influenced the quality of the trials since most of the authors rushed their publications, which might have led to low-quality publications.

### 3.2. Chloroquine/Hydroxychloroquine (CQ/HCQ)

CQ and its analog HCQ are drugs derived from 4-aminoquinolones. The dosage of CQ ranges from 50- to 150-mg, while that of HCQ is 400-mg, and residues of these drugs can last for weeks or months in the body [64]. CQ was first developed in the 1940s for the treatment of malaria. It is indicated for preventing and treating an acute attack of malaria caused by *Plasmodium vivax*, *Plasmodium ovale,* and *Plasmodium malarie*. HCQ is a racemic mixture consisting of an *R* and *S* enantiomer. It is commonly prescribed to treat malaria, rheumatoid arthritis, chronic discoid lupus erythematosus, and systemic lupus erythematosus [64]. However, since the beginning of the COVID-19 pandemic, several studies evaluated its effect against SARS-CoV-2 infection due to its antiviral properties against several other viruses, such as the Influenza virus and SARS-CoV [65,66,67].

The mechanism of action of HCQ is exerted on the membranous structures of *Plasmodium*, which causes the lysis and death of the parasite. In *P. malariae* and *P. vivax* malaria, it is effective in stopping acute attacks and prolonging the intervals between treatment and relapse. In *Plasmodium falciparum* malaria, it prevents the acute attack and can lead to the cure of the disease, except in the presence of resistant strains [68].

#### 3.2.1. Antiviral Mechanism of the Chloroquine/Hydroxychloroquine (CQ/HCQ)

Even though very similar, HCQ and CQ have some differences regarding their antiviral mechanism. For instance, CQ is responsible for neutralizing the pH of the lysosome, which could prevent vital viral pathways, such as the S protein cleavages, and make it difficult for the virus to enter host cells. CQ is also responsible for the inhibition of the lysosomes and autophagosomes, mainly due to Syntaxin 17 dysregulation, ultimately leading to blockage of lysosome transportation, mainly by affecting the Golgi apparatus [69,70,71]. In addition to the CQ effects, HCQ inhibits the movement of SARS-CoV-2 from early endosomes to early lysosomes, thus further disrupting the release of viral genetic material [70,72], a similar mechanism observed for Ivermectin.

For successful entry into a host cell, SARS-CoV-2 relies strongly on the interaction between its Spike protein and the Angiotensin-Converting Enzyme (ACE)-2 protein from the host cell, which is necessary for SARS-CoV-2 entry into the cell [73,74]. CQ reduces ACE-2 glycosylation, inhibiting this type of interaction and, therefore, prevents the virus from binding and integrating into new cells [75,76,77]. On the other hand, HCQ might inhibit the interaction between viral Spike protein and the cell membrane, mainly binding in gangliosides [75,77]. In addition, CQ and HCQ also inhibit glycosyl-transferases, post-translational viral modification, quinone reductase-2, sialic acid synthesis, and replicative viral mechanisms [77,78].

Endothelial adhesion molecules (Selectins, Intercellular Adhesion Molecule 1, and Vascular Cell Adhesion Molecule 1), pro-inflammatory cytokines (IL-6, IL-2 receptor, and TNF-α), and pro-inflammatory chemokines (monocyte chemoattractant protein-1) are mediators in COVID-induced endothelial dysfunction [79]. Since CQ/HCQ targets several pro-inflammatory cytokines, another possible mechanism is the anti-inflammatory effect, which might be effective against COVID-19 since SARS-CoV-2 can produce a cytokine storm, especially in the second phase of infection [79,80].

HCQ might also interfere in the cytokine storm caused by COVID-19 through several pathways, such as the inhibition of the presentation of antigen by the antigen-presenting cells to T cells, declining the total of T cells activations, the blockage of TLR-9, activation of TLR-7, interfering in the Stimulator of Interferon Gene (*STING*) pathway—cyclic guanosine monophosphate–adenosine monophosphate (GMP-AMP) synthase, and ultimately decreasing the production of cytokines [70,81,82,83,84].

The processing of antigens to peptides in Antigen Presenting Cells (APCs) is disrupted by HCQ, which, in turn, disrupts peptide presentation for major histocompatibility complex class II (MHC-II) cells. The disruption caused by HCQ also interferes with B-cell activation by CD4+ (cluster of differentiation 4) T-cells; thus, this diminishes their functions and cytokine production [IL-1, IL-6, Interferon (INF)-gamma (INF-γ), TNF-α, and B-cell activating factor] [77].

The following mechanisms can perform the anti-inflammatory action of CQ: suppression of T lymphocyte responses to mitogens, inhibition of leukocyte chemotaxis, stabilization of lysosomal enzymes, processing through the Fc receptor, inhibition of DNA synthesis, and RNA and free radical scavenging [67]. HCQ, in inflammatory conditions, blocks TLR-9 on dendritic cells, which recognizes immune complexes that contain DNA, leading to the production of interferon and the maturation of dendritic cells that present the antigen to T cells. Thus, its blockade reduces the activation of dendritic cells and inflammatory processes [77].

Furthermore, by increasing the endosome and lysosome pH of dendritic cells, intracellular antigen processing, and peptide loading into MHC-II molecules are suppressed, reducing T-cell activation [67,85]. Thus, these drugs can be used for autoimmune disorders such as rheumatoid arthritis and systemic lupus erythematosus, with HCQ being preferred in these cases, not only because of its mechanism of action but also because of its lower toxicity [67]. The antiviral mechanism is presented in Figure 3.

#### 3.2.2. Efficacy of Chloroquine/Hydroxychloroquine (CQ/HCQ) to Treat Coronavirus Disease (COVID)-19 in Randomized Controlled Trials (RCTs)

The majority of studies found no significant benefit of CQ/HCQ in reducing mortality or severity in patients with COVID-19. In this context, in the systematic review, we obtained a total of 1715 studies using the descriptors as we described above. From those studies, 323 were excluded for being duplicates. We also excluded 1381 studies that did not meet the inclusion criteria, as described in Figure 4. A total of 11 studies met our inclusion criteria (Figure 4) [19,25,26,27,28,29,30,31,32,33,34]. In Table 3, we assessed all available phase III RCTs that evaluated HCQ or CQ as a treatment against COVID-19.

The RECOVERY trial had a sample of 4716 participants who underwent randomization, and it was designed as a platform trial and involved 176 hospitals in the United Kingdom (UK) [25]. To evaluate the HCQ efficacy in treating participants with COVID-19, 1561 participants received HCQ, and 3155 received usual care (not informed in the RCT) [25]. The remaining participants were assigned to different treatment groups [25]. Those individuals were treated with four tablets of 200-mg of HCQ Sulfate at baseline and 6-h, followed by two tablets starting at 12-h after the initial dose and then every 12-h for the following nine days or until discharge, whichever occurred earlier [25]. Mortality on the 28th day was assessed as the primary outcome. In the HCQ group, 27% of the sample perished versus 25% in the usual care group, which allows the authors to conclude that such a group did not have a lower incidence of death [25].

Self et al. (2020) also analyzed hospitalized individuals with positive RT-PCR for SARS-CoV-2 [26]. This multicenter study comprised 479 randomized participants, 242 assigned to HCQ and 237 to placebo [26]. The participants in the HCQ received 400-mg of HCQ twice a day for the first two doses and then 200-mg twice a day for eight doses [26]. The primary outcome was the clinical status on day 14, evaluated by an ordinal scale [26]. On the 14th day, the mean interquartile of the ordinal scale for the HCQ and placebo group showed no significant statistical difference (six versus six, respectively) [26].

A multi-center phase III clinical trial conducted by Dubée et al. (2021) analyzed a sample of 250 participants, HCQ (n = 110) and placebo (n = 116) [29]. On the first day, the HCQ group received four tablets of 200-mg (total 800-mg) and, for the following eight days, two tablets of 200-mg (total of 3200-mg) a day, four grams of HCQ in total [29]. Death and the need for invasive mechanical ventilation within the 14 days following randomization were assessed as primary outcomes [29]. However, until the study’s discontinuation, no benefit of HCQ therapy during RT-PCR positivity could be evaluated. Therefore, the authors conclude that no significant results can be drawn on the efficacy of HCQ [29].

Results from the SOLIDARITY Consortium trial, published by Pan et al. (2021), promoted by the WHO showed that HCQ has no definite effect on COVID-19 mortality [31]. A total of 1863 participants were randomized: 954 were assigned to receive HCQ, and 909 did not receive the drug [31]. The SOLIDARITY Consortium trial was a multiarmed RCT that evaluated several other drugs, such as Remdesivir, Lopinavir, and Interferon [31]. Those participants in the HCQ group received four tablets (800-mg) of HCQ sulfate at hour zero, four tablets at hour six, and starting at hour 12—two tablets (400-mg) twice daily for 10 days [31]. The primary outcome was to evaluate death within 28 days [31]. Death occurred in 104 of 947 participants in the HCQ group and 84 of 906 participants in the control group, with no statistical difference [31]. 

A multi-center study conducted in Mexico by Hernandez-Cardenas et al. (2021) randomized 214 participants with COVID-19, confirmed by RT-PCR, <14 days of symptom onset, and lung injury requiring hospitalization [30]. The primary outcome was 30-day mortality [30]. In the trial, 106 participants received 200-mg HCQ every 12-h for 10 days, and 108 received the placebo [30]. In the follow-up, 30 days after randomization, the mortality rate was similar in both groups (38% versus 41% for the HCQ and placebo group, respectively). Thus, no benefit or harm from HCQ usage was demonstrated [30]

A multicenter study conducted by Ader et al. (2021) analyzed 603 hospitalized individuals with positive SARS-CoV-2 RT-PCR that used Lopivanir/Ritonavir and Lopinavir/Ritonavir plus Interferon-β-1a and compared these treatments with HCQ and standard of care treatments to evaluate clinical status on day 15 measured on the WHO scale [27]. Participants received 400-mg Lopinavir and 100-mg Ritonavir twice a day for 14 days plus 44 μg Interferon-β-1a on days one, three, and six and 400-mg HCQ twice on day one and 400-mg daily for nine days [27]. The study found that HCQ did not improve clinical status compared to the control group [27].

Arabi et al. (2021) evaluated 726 individuals with confirmed or suspected COVID-19 and who were also receiving respiratory or cardiovascular organ failure [28]. The trial accounted for four arms: (i) 268 participants were assigned to Lopinavir/Ritonavir (249 included in the final analysis); (ii) 52 participants were assigned to HCQ (49 included in the final analysis); (iii) 29 participants were assigned to Lopinavir/Ritonavir plus HCQ (26 included in the final analysis), and (iv) 377 participants were assigned to receive no antiviral (353 included in the final analysis) [28]. The primary outcome was the ordinal scale of the number of respiratory and cardiovascular organ support-free days and hospital mortality. The median organ support-free days among participants in Lopinavir-Ritonavir, HCQ, and combination therapy groups were 4 (−1 to 15) [Odds Ratio (OR)—95% confidence interval (95%CI) = 0.73 (0.55 to 0.99)], 0 (−1 to 9) [OR (95%CI) = 0.57 (0.35 to 0.83)], and −1 (−1 to 7) [OR (95%CI) = 0.41 (0.24 to 0.72)], respectively, compared to 6 (-1 to 16) days in the control group. In-hospital mortality among participants in Lopinavir-Ritonavir, HCQ, and combination therapy was 88/249 (35.3%) [OR (95%CI) = 0.65 (0.45 to 0.95)], 17/49 (34.7%) [OR (95%CI) = 0.56 (0.36 to 0.89)], and 13/26 (50%) [OR (95%CI) = 0.36 (0.17 to 0.73)], respectively, compared to 106/353 (30%) in the control group. In critically ill patients the intervention resulted in worse outcomes [28].

The final analysis of the multicenter study performed by Ader et al. (2022) enrolled 603 COVID-19 inpatients who required oxygen or ventilatory support [33]. The study evaluated the patient’s clinical status at day 15 measured by the WHO scale comparing the standard of care, the standard of care plus Lopinavir/Ritonavir, the standard of care plus Lopinavir/Ritonavir plus Interferon-β-1a and the standard of care plus HCQ [33]. Participants from the standard of care plus Lopinavir/Ritonavir group received 400-mg Lopinavir and 100-mg Ritonavir orally twice a day for 14 days [33]. Participants from the standard of care plus Lopinavir/Ritonavir plus Interferon-β-1a group received the same plus 44-mg subcutaneous Interferon-β-1a on days one, three, and six [33]. Participants from the standard of care plus HCQ groups received 400-mg orally, twice on day one as a loading dose followed by 400-mg once daily for nine days [33]. The study found that Lopinavir/Ritonavir, Lopinavir/Ritonavir plus Interferon-β-1a and HCQ were not associated with clinical improvement at day 15 and day 29, nor in a reduction in viral shedding, and generated significantly more severe adverse effects in Lopinavir/Ritonavir-containing arms [33].

In a phase III RCT that took place in Mexico, Beltran Gonzalez et al. (2022) evaluated a total of 106 hospitalized individuals with confirmed or suspected COVID-19 pneumonia [19]. Out of the 106 individuals, 33 participants received HCQ, while 36 individuals received 12-mg (<80 kg) to 18-mg (>80 kg) of Ivermectin according to patients’ weight and 37 individuals received only a placebo [19]. It is worth mentioning that due to the RECOVERY trial [60], all the individuals who needed oxygen support received dexamethasone 6-mg intravenous once per day for 10 days or until discharged during the last week of June 2020 [19]. The primary outcome was a composite of the length of hospital stay, death, and respiratory deterioration [19]. No difference in hospital stay was observed in the Ivermectin group (seven days; IQR = 3 to 9 days) compared to placebo (five days; IQR = 4 to 7 days), nor in respiratory deterioration or death [6/33 (18.1%) versus 9/37 (24.3%)] nor death alone [2/33 (6%) versus 6/37 (16.2%)] [19]. There was no benefit in using HCQ to treat hospitalized individuals with confirmed or suspected COVID-19 [19].

A Brazilian study conducted by Rea-Neto et al. (2021) evaluated individuals who were admitted to intensive care units or acute care rooms with flu symptoms and dyspnea or need for supplemental oxygen or SpO_2_ < 94 on room air or computed tomography scan compatible with COVID-19 or need for mechanical ventilation and a confirmed diagnosis of COVID-19 [32]. A total of 142 individuals were randomized, however, only 105 subjects were analyzed in the modified intention-to-treat analysis [32]. In the study, 53 participants were allocated to the intervention group and received CQ 450-mg twice a day on day one followed by 450-mg once a day from days two to five or HCQ 400-mg twice a day followed by 400-mg once daily from days two to five plus standard of care whereas 52 participants received placebo plus standard of care [32]. The primary outcome was the clinical status on day 14 with a 9-point ordinal scale [32]. On the 14th day, the odds of having an unfavorable clinical outcome were higher in the CQ/HCQ group, even after controlling for confounding factors [OR (95%CI) = 2.45 (1.17 to 4.93)]. On the 28th day, individuals in the CQ/HCQ group also presented worse clinical outcomes (OR = 2.47 (95%CI = 1.15 to 5.30). The mortality rate on the 28th day was not different between groups [Relative risk (RR) (95%CI) = 1.57 (0.79 to 3.13)] [32]. The authors concluded that CQ/HCQ treatment may be associated with worse clinical status, increased risk of renal dysfunction, and need for mechanical ventilation [32].

Finally, the trial NCT04358081 randomized 20 hospitalized individuals with confirmed COVID-19 in the USA, however, only 19 individuals were evaluated since one of them was mis-randomized [34]. A total of seven participants received HCQ 600-mg once a day followed by 200-mg three times a day plus an Azithromycin placebo, seven participants received HCQ 600-mg once a day followed by 200-mg three times a day plus a placebo, and five participants received only a placebo [34]. The primary outcome accounted for clinical response by day 15 defined as discharged alive or no need for mechanical ventilation or no need for supplementary oxygen therapy [34]. On the 15th day, 7/7 (100%) individuals achieved clinical improvement in both groups I (HCQ plus placebo) and II (HCQ plus Azithromycin), whereas in group III, 4/5 (80%) (placebo) achieved clinical improvement [34]. All the participants treated with HCQ with or without Azithromycin presented clinical improvement on the 15th day compared to placebo [34].

In most of the trials, the use of CQ/HCQ did not improve the severity nor mortality among hospitalized patients with COVID-19. Further systematic reviews with meta-analysis did not also observe better outcomes in those using CQ/HCQ [86,87].

Most of the studies presented a low risk of bias (8/11), and only three studies were classified as high risk/presented some concerns. The results of the risk of bias are shown in Table 4.

### 3.3. Azithromycin

Azithromycin is an antimicrobial drug from the macrolide group and chemically consists of a macrocyclic lactone ring, which binds to one or more sugars. In addition to Azithromycin, Clarithromycin, Erythromycin, Spiramycin, Miocamycin, Roxithromycin, and other antibiotics, belong to this group of drugs. The spectrum of action between these drugs is similar, differing only in potency against some microorganisms since the structure of Azithromycin has a nitrogen atom in the lactone ring [88,89]. Thus, this rearrangement increased the drug’s spectrum of activity, providing an increased tissue level higher than the serum level and a prolonged tissue half-life that allows for dose reduction during treatment [90].

Azithromycin has a broad spectrum of action showing activity against gram-positive bacteria and a range of gram-negative bacteria. Therefore, Azithromycin is less active than other members of the macrolide class against Gram-positive microorganisms (*Streptococcus* spp. and *Enterococci*). However, it is very active against *Moraxella catarrhalis*, *Pasteurella multocida*, *Chlamydia* spp., *Mycoplasma pneumoniae*, *Legionella pneumophila*, *Borrelia burgdorferi*, *Fusobacterium* spp., and *Neisseria gonorrhoeae*. It is also an excellent drug choice against some protozoa, such as in the treatment of *Leishmania amazonenses* infection [88,90].

Azithromycin has less gastric intolerance than other drugs of the same family, such as Erythromycin, and has a longer half-life, making it possible to use a single dose [91]. In addition to having good tissue penetration, the drug accumulates inside some cells, mainly macrophages [92]. The antimicrobial reaches adequate concentrations in aqueous humor, middle ear, screw sinuses, nasal mucosa, tonsils, lung tissue, pleura, kidneys, liver, bile ducts, skin, and prostate. On the other hand, it does not have good penetration into meninges, bone tissue, and synovial fluid. Azithromycin is previously eliminated via the liver, and only a small amount is found in the urine [88,93].

Macrolides are a therapeutic alternative for Penicillin-allergic patients and can be used in the following conditions: (i) respiratory tract infections by group A *Streptococcus*, (ii) pneumonia by *Streptococcus pneumoniae*, (iii) prevention of endocarditis after a dental procedure, (iv) superficial skin infections by *Streptococcus pyogenes*, (v) prophylaxis of rheumatic fever (*Streptococcal pharyngitis*), and (vi) considered the first choice in the treatment of pneumonia caused by atypical bacteria (*M. pneumoniae*, *L. pneumophila*, and *Chlamydia* spp). Azithromycin can be used for the treatment or prevention of infections by *Mycobacterium avium*-intracellular, *Helicobacter pylori*, *Cryptosporidium parvum*, *Bartonella henselae* (bacillary angiomatosis, common in patients with human immunodeficiency virus—HIV), Lyme disease, and *Toxoplasma gondii*. The drug has schizonticidal activity against *Plasmodium* spp. and can be used as prophylaxis against CQ-resistant *P. falciparum* [94].

#### 3.3.1. Antiviral Mechanism of Action of the Azithromycin

Macrolides are bacteriostatic agents that inhibit protein synthesis through their reversible binding to the 50 S ribosomal subunit of sensitive microorganisms, preventing messenger RNA (mRNA) translation without affecting nucleic acid synthesis and reducing bacterial biofilm production and quorum-sensing (how the bacterium controls the expression of genes that may be favorable for a given situation, enabling the bypassing of environmental barriers) [95].

The appearance of pro-inflammatory cytokine storms characterizes respiratory viral infections. Some in vitro and in vivo studies established that viruses are potent inducers of several cytokines and chemokines, including TNF-α, IFN-γ, IFN-α/β, interleukins (IL-6 and IL-1), macrophage inflammatory protein-1 (MIP-1), among others [96,97]. In this way, macrolides seem to negatively regulate the inflammatory cascade, attenuating the excessive production of cytokines in viral infections. In addition, these drugs can influence phagocytic activity by modifying several functions, including chemotaxis, phagocytosis, oxidative burst, bacterial killing, and cytokine production [98]. It has also been reported that this class of drugs can interfere with the Influenza virus replication cycle, resulting in the inhibition of virus production from infected cells, primarily by inhibiting intracellular hemagglutinin HA0 proteolysis [97,99].

In vitro studies have shown that Azithromycin, widely used in individuals with COVID-19, could have antiviral effects on bronchial epithelial cells and was also shown to be immunomodulatory and reduce exacerbations in chronic airway diseases [100,101]. Another hypothesis would be that Azithromycin could interfere with the binding between the SARS-CoV-2 Spike protein and the ACE-2 receptor protein from the host cell, preventing the virus from entering the cell [102].

However, Azithromycin may increase the risk of life-threatening ventricular arrhythmias or cardiac arrest due to corrected QT (QTc) interval prolongation [103] and this risk may be increased in the presence of other drugs known to prolong the QTc interval, such as HCQ [104]. Thus, evidence on the efficacy and safety of adding Azithromycin to the treatment regimen for COVID-19 is limited by mostly non-randomized and low-quality studies [105,106]. In this context, the antiviral mechanism of action for the Azithromycin against SARS-CoV-2 is presented in Figure 5. 

#### 3.3.2. Efficacy of Azithromycin to Treat Coronavirus Disease (COVID)-19 in Randomized Controlled Trials (RCTs)

The majority of studies found no significant benefit of Azithromycin in reducing mortality or severity in patients with COVID-19. In this context, in the systematic review, we obtained a total of 808 studies using the descriptors we described above. From those studies, 111 were excluded for being duplicates, and 694 were excluded for not meeting the inclusion criteria as described in Figure 6. A total of three studies met our inclusion criteria (Figure 6) [35,36,37]. In Table 5, we assessed all available phase III RCTs which evaluated Azithromycin as a treatment against COVID-19.

A Brazilian trial conducted by Cavalcanti et al. (2020) randomized a total of 667 participants with a positive RT-PCR test for SARS-CoV-2, +50 years old with at least one comorbidity who also presented mild to moderate COVID-19 [35]. This multicenter open-label trial has already been aforementioned; however, the authors concluded that HCQ with or without Azithromycin should not be used in patients with mild to moderate COVID-19 [35].

The COALITION II study evaluated whether adding Azithromycin to the standard of care, including HCQ, would improve clinical outcomes for individuals admitted to the hospital with severe COVID-19 [36]. In an open-label RCT in 57 centers in Brazil, hospitalized individuals with suspected or confirmed COVID-19 and at least one additional severity criterion were enrolled [36]. The inclusion criteria were the use of oxygen supplementation greater than 4 L/min or a high-flow nasal cannula and non-invasive mechanical ventilation or invasive mechanical ventilation [36]. The study enrolled 447 adult participants at several hospitals in Brazil, approximately one-third of whom were women [36]. Participants were randomly assigned (1:1) to Azithromycin (500-mg orally, nasogastric, or once daily intravenous administration for 10 days) plus complete standard of care (which included the use of HCQ) or standard of care without macrolides [36]. All participants received HCQ (400-mg twice daily for 10 days) because it was part of standard care in Brazil for individuals with severe COVID-19 [36]. The standard of care was based on local guidelines [36]. The primary outcome was the clinical status on 15 days, assessed using a six-level ordinal scale ranging from non-hospitalization to death [36]. Participants were followed for 29 days to assess 29-day mortality [36]. The primary outcome was evaluated in all intent-to-treat participants who had SARS-CoV-2 with molecular or serological testing before randomization [36]. The study found no benefit of Azithromycin on clinical outcomes, including clinical status or mortality, when added to the standard treatment regime and no evidence of increased adverse events with the addition of Azithromycin [36]. There was no evidence of a difference in outcomes by sex, although a pre-specified subgroup analysis suggested potentially worse clinical status on 15 days in younger individuals receiving Azithromycin [36].

In another randomized, controlled, open-label, adaptive platform trial, several possible treatments (Dexamethasone, CQ, and Lopinavir-Ritonavir) were compared with usual care in participants hospitalized with COVID-19 in the UK [37]. Eligible and consenting participants were randomly allocated to the usual standard of care alone or usual standard of care plus Azithromycin 500-mg once daily orally or intravenously for 10 days or until discharge [37]. The standard of care followed the guidelines of the hospital where the participants were treated [37]. The participants were assigned via simple randomization (non-stratified) web-based with allocation concealment and were twice as likely to be randomly assigned to usual care than to any of the active treatment groups [37]. Participants and study site staff were not masked to their allocated treatment, but all others involved were masked to outcome data during the study [37]. The primary endpoint was all-cause mortality at 28 days, assessed in the intent-to-treat population [37]. In the study, 561 (22%) participants in the Azithromycin group and 1162 (22%) participants assigned to the usual care group died within 28 days [37]. No significant differences were observed in the length of hospital stay or the proportion of individuals discharged from the hospital alive within 28 days [37]. Among those not on invasive mechanical ventilation at baseline, no significant difference was observed in the proportion who met the composite endpoint of invasive mechanical ventilation or death [37]. Thus, in hospitalized participants with COVID-19, Azithromycin did not improve survival or other pre-specified clinical outcomes. Azithromycin use in hospitalized individuals with COVID-19 should only be restricted to individuals with a clear antimicrobial indication [37].

Thus, the RCTs found in the literature do not justify the routine use of Azithromycin to reduce recovery time or the risk of hospitalization of people with suspected COVID-19 or reduce the risk of hospitalization and subsequent death [35,36,37]. In the same way, in a recent meta-analysis from the Cochrane Database, which included 11 RCTs, it was observed Azithromycin did not improve 28-day mortality or had a significant clinical improvement on day 28 of hospitalization [107]. In addition, the widespread use of antibiotics for the treatment and prevention of viral infection could lead to future consequences, such as an increase in antimicrobial resistance that could lead to infections by multidrug-resistant or even pan-resistant bacteria soon [108].

Only one of the studies presented a low risk of bias, the other two were classified as having some concerns. The data is shown in Table 6.

### 3.4. Impact of the Use of Unproven Scientific Drugs to Treat Coronavirus Disease (COVID)-19 in Brazil and the World

The COVID-19 pandemic hit the world unprecedently since it was a disease caused by a novel virus with no known treatment. Several already known substances were proposed as treatments, such as CQ/HCQ, Ivermectin, and Azithromycin [6,10,11], which was coined the name of repurposed drugs, that is, to use an already known substance for the treatment of some condition it was not firstly developed for [13]. Even though repurposed drugs are relatively common, and there are even successful examples in the literature, such as the use of sodium-glucose transport protein-2 (SGLT2) inhibitors that were firstly developed for diabetes mellitus [109], but also showed benefit in patients with heart failure chronic kidney disease [110,111], its success is limited [14,112]. Hill’s specificity principle is one of the main principles for the use of repurposed medicines. Generally, medications that are not specific for a given disease only rarely have a satisfactory effect size, conversely, therapies that are more specific tend to be more effective and have satisfactory effect size [14].

In addition, in Brazil, the government presented many errors in the management of the COVID-19 pandemic, including, for example, the difficulty in managing the vaccination against the SARS-CoV-2, the availably of intensive care unit beds, and the purchase of drugs to manage, the most severely affected individuals [6,7,8,113,114,115,116,117]. Moreover, in Brazil, the COVID-19 pandemic was associated with high indexes of underreporting and disparities among Brazilian people as presented by several epidemiological studies [118,119,120,121,122,123,124,125]. Curiously, in Brazil, we did not learn our lesson regarding COVID-19 management, as we observed in the follow-up of the first cases of MonkeyPox and a higher incidence of Dengue fever [126,127] with a concomitant forgetfulness about the impact that COVID-19 caused in Brazil [128].

Due to the absence of an efficacy treatment and the high number of deaths from COVID-19, scientists, physicians, and politicians rush to find a new possible treatment. Perhaps, for physicians, this seeking for a viable treatment might have been even more intense since many of them think “We must give a drug to treat the patient” [14]. This scenario encouraged the adoption of unproven drugs to treat COVID-19, like Ivermectin, CQ/HCQ, and Azithromycin, with no prior RCT, which proves its effectiveness [14]. Many politicians, such as the Brazilian president, and the former American president, publicly endorsed these drugs as silver bullets in the COVID-19 treatment [6,129,130]. Unfortunately, this “infusion of politics into science”, which the international scientific community advised against, occurred worldwide, which might have contributed to the widespread use of these drugs before the publication of RCTs [129,130,131].

Specifically in Brazil, the sales of the drugs part of the so-called “COVID-Kit”, such as Ivermectin, CQ/HCQ, and Azithromycin, enhanced exponentially. Nearly one in four individuals had taken these drugs [132] which might have contributed to a shortage of these drugs for patients who use them on-label. For instance, in Brazil, HCQ treats rheumatic diseases like lupus or rheumatoid arthritis. In June 2020, pharmacies all over Brazil had a shortage of this drug due to the increased sales to treat COVID-19, which impaired the treatment of rheumatic patients [130]. In the same way, a report in Brazil showed that several individuals who had a prior COVID-19 diagnosis also made use of drugs from the “COVID-19-Kit” being that 19% of them took CQ/HCQ, 55% took Ivermectin, and most of them, that is 77%, took Azithromycin [133]. Not only do these drugs have no efficacy against COVID-19, but they give a “safety” feeling, in which individuals who take them think they are protected against the SARS-CoV-2 infection, leading them to relax measures that are proven to decrease SARS-CoV-2 viral spread, such as social distancing and use of masks, which might have been observed in the DETECT-CoV-2 Brazilian report, that showed a 50% increase of infection in individuals who made use of the COVID-19-kit as prophylactic measures [130,133].

Several international scientific entities, such as the National Institute of Health, and the Infectious Disease Society of America, advise against the use of the COVID-19-Kit outside clinical trials, and yet, the Brazilian Ministry of Health insists on advocating the use of these drugs to treat COVID-19, with the release of a technical note encouraging the use of HCQ over vaccines against COVID-19 [134,135]. Interestingly, many Brazilian associations, such as the Brazilian Medical Association and the Brazilian Societies of Infectious Diseases and Pulmonology which are not associated with the federal government and, in this case, exempt from political bias, are following the international scientific community, recommending against the use of the Brazilian COVID-19 kit [130]. Noteworthy, even after the publication of dozens of meta-analyses and robust clinical trials showing no efficacy of CQ/HCQ, Ivermectin, and Azithromycin to treat COVID-19 [62,86,107] politicians still have encouraged the use of these drugs to treat COVID-19. One might speculate reasons that led them to do so, for instance, the fear of being wrong and actively having fought for drugs that in reality has no effect; not understanding scientific thinking, which can make politicians trust a methodologically flawed study that has a positive result instead of a methodologically well-done study with a negative result; and the fact many physicians, who also have limited knowledge of evidence-based medicine, continue to prescribe these medications.

In Brazil, there has been a reduction in vaccination coverage, mainly in pediatric groups [136,137]. Several factors can be linked to this reduction, including anti-vaccine movements by the community and politicians [6,7,114,138,139]. During the COVID-19 pandemic, this issue was highlighted, with disbelief in vaccines against COVID-19 being described, mainly due to political polarization. Along the same path, there was an intense political movement associated with reducing effective contagion measures against COVID-19, including social isolation and the use of facial masks [6,7,8]. Furthermore, the intense spread of news (especially fake news), linked to the federal government that highlighted the use of drugs, not effective against COVID-19 was described. Among them, CQ/HCQ and Ivermectin stood out. In view of the above, the political movement in Brazil, during COVID-19, highlighted two crucial points: (a) disbelief in science and (b) party political propaganda with the aim of disrupting the management of the pandemic by bringing tools that could normalize daily activities of society with a focus on the economy through the use of the social media and false reports [6,7,114,115,130,140,141,142,143]. Among the reports, the politicians inferred that COVID-19, in reality, was just a weak flu, that the vaccine against COVID-19 would be responsible for turning the population into crocodiles, vaccines are not safe and cause illnesses, for example, COVID-19 itself or autism, the vaccine contains human tracking chips and are used for population control, the vaccine is made from human fetuses and vaccines contain Luciferase referring to Lucifer. At the same time, the aforementioned drugs were declared to protect against viral infection and to be effective in-patient management. These statements led to the indiscriminate use of drugs and a reduction in their availability to individuals who really needed them. The assessment of the purpose of such propagation of incorrect information is complex, however, in the authors’ point of view, it is linked to a lack of scientific-technical knowledge, the generation of false hope with the aim of providing a return to economic activities, political polarization to promote gain of votes, and use of bad faith in managing the opinions of people with a low level of scientific knowledge.

#### 3.4.1. Impact of the Use of Ivermectin to Treat Coronavirus Disease (COVID)-19

The economic impact of Ivermectin was enormous in the world, especially in the USA and Latin America. In the USA, a study evaluated how much the private and Medicare insurance programs spent on Ivermectin prescriptions for COVID-19 from 1 December 2020 to 31 March 2021, and the private insurance dispensed a total of 4700 prescriptions and 891 by Medicare. However, Ivermectin is relatively cheap; both these insurances spent nearly $13 million in a year on a drug that does have any effect on COVID-19 [144], which could have been used for drugs that decrease COVID-19 mortality, such as Dexamethasone [60], or even in critical care, to invest in ventilators.

In the same way, Brazilian sales of Ivermectin increased from 1.5 million units in December 2021 to 5.5 million units in January 2022, mainly due to the advance of the Omicron variant [145], which corresponds to nearly 22 million dollars spent by the Brazilian citizens on a drug that has no efficacy against COVID-19. It is understandable why many lay individuals believe in this drug, both in Brazil and the USA, since two organizations, namely Physicians for Life in Brazil and Front Line COVID-19 Critical Care in the USA, advocate for the use of Ivermectin. Even World Ivermectin Day was promoted by these organizations, in which they celebrated the use of Ivermectin to treat COVID-19. Perhaps this enterprise was promoted by the desire to be free from the virus. However, we cannot exclude these scientists’ interest in prestige [130]. Unfortunately, it is also understandable why so many physicians are prescribing this drug for COVID-19 since several websites have compiled and conducted a meta-analysis on the evidence of Ivermectin and COVID-19, such as https://ivmmeta.com/ (accessed on 22 August 2024) and https://c19ivermectin.com/ (accessed on 22 August 2024), claimed that Ivermectin could decrease the need for mechanical ventilation, hospitalization, and death. However, several methodological flaws were reported as the websites did not make the protocol registration available in the methods section, inclusion criteria, quality assessment, and search strategies, which undermined the reliability of the results of this website [58]. Perhaps, due to how these results are presented on the websites and the fact many physicians have a common understanding of basic research principles [146], they were more inclined to prescribe Ivermectin.

Another justification for using Ivermectin in COVID-19 is its safety profile. Although relatively safe, with mild adverse effects, such as diarrhea, dizziness, abdominal pain, and vomiting, more serious adverse effects were reported, such as lethargy, seizures, and coma in those who had taken supra dosages or with “leaky” blood-brain barrier [45,46,47]. Due to the high intake of Ivermectin, several more severe adverse effects were described; in Brazil, 12 cases in three months of Ivermectin-related hepatitis were reported, even with an increased need for liver transplant [47]. In the same way, in the Oregon Poison Center (USA), calls related to Ivermectin intoxication increased from 0.25 calls per month in 2020 to 0.86 calls per month between January and July 2021, being that the individuals reported they were using Ivermectin as a treatment or even as prophylaxis for COVID-19. Of the 21 individuals who called the Oregon Poison Center in August, six needed hospitalization, and four needed intensive care unit treatment due to Ivermectin intoxication [147]. Even with these intoxication reports, individuals are organizing “buyers’ clubs”, especially in the UK, to import Ivermectin to treat COVID-19 [148]. One might believe Ivermectin may be a new “snake oil” [149] that is supposed to treat and cure COVID-19 with few adverse effects. However, good methodological trials and real-world observation data say otherwise. Thus, health institutions and governments should discourage the off-label use of Ivermectin to treat patients with COVID-19.

Finally, it is important to evaluate the literature with caution. For example, one published meta-analysis [150] suggested a benefit of Ivermectin. However, several methodological concerns have been noted regarding the meta-analysis performed by Bryant et al. (2021) [150], for instance, 13 out of 24 of the included papers have at least one high risk of bias in at least one domain. Most of the studies are heterogenous between each other, although they evaluate the same outcomes (for example, mortality), and the sample are very different in each study (for example, one study evaluated outpatients with mild COVID-19, another one evaluated hospitalized with severe COVID-19) which could have biased the analysis. Not only that, recently, it was published an expression of concern has been issued for this meta-analysis [151]. The note claims that the authors collected inaccurate data and/or reported in at least two primary sources of the meta-analysis. Finally, a more robust meta-analysis, such as the one performed by the Cochrane group, did not observe any benefit of using Ivermectin to treat COVID-19 [62].

#### 3.4.2. Impact of the Use of Chloroquine/Hydroxychloroquine (CQ/HCQ) to Treat Coronavirus Disease (COVID)-19

Perhaps, CQ/HCQ was the first treatment hype for the SARS-CoV-2 infection during the COVID-19 pandemic since the flawed study by Gautret et al. (2020) [152]. However, several methodological issues were observed, such as lack of randomization, exclusion of four patients with worst outcomes from the analysis (one patient died, and three patients needed intensive care unit), and non-blindness, which could hamper their conclusion of the efficacy of CQ/HCQ to decrease the SARS-CoV-2 viral load [153,154]. Unfortunately, this flawed study was responsible for mass media coverage of CQ/HCQ and was even endorsed by several authorities [6,155]. The endorsement of unproven drugs to treat COVID-19 is dangerous since it was responsible for increasing the searches on Google about HCQ [156], especially on the day the former USA president endorsed the drug, and also because in the early COVID-19 pandemic, the lay population was not able to discern correctly whether an HCQ was effective or not [157].

After the public endorsement by such important figures, the sales of CQ/HCQ skyrocketed, especially in the USA and Designated Market Areas that had more votes for Mr. Donald Trump [158,159], which might have contributed to a shortage of CQ/HCQ [160]. As aforementioned, these drugs are used mainly to treat rheumatologic diseases, such as lupus erythematosus, rheumatoid arthritis, and malaria, the lack of use by rheumatologic patients by the shortage of these drugs might prejudice their treatment as discontinuation might result in a clinical flare-up [160,161,162]. Furthermore, there are even reports of individuals who invertedly self-medicated themselves with CQ/HCQ to treat COVID-19 [163]; however, self-medication per se is dangerous for the patients since they do not know the side effects, toxicity, and indications of correct use of the drugs, thus contributing to side effects, as hepatotoxicity and QTc prolongation, harming the patients even further [35,164].

The polarization of whether individuals should use CQ/HCQ in the early pandemic was unprecedently, and it contributed to many scientists’ attacks worldwide [165]. Several researchers reported harassment in the form of attacks on credibility, reputation damage, and even death threats [165], as was the case of Mr. Marcus Lacerda, the leading investigator of a Brazilian clinical trial that observed HCQ in high doses to be ineffective against COVID-19 [166,167].

#### 3.4.3. Impact of the Use of Azithromycin to Treat Coronavirus Disease (COVID)-19

Since Azithromycin is a macrolide antibiotic, it is commonly used to treat bacterial infections caused by *Streptococcus*, *Legionella*, *Mycoplasma*, and *Chlamydia* [94]. Although an antiviral mechanism was described in vitro [100,102], this exact mechanism is not observed in the RCTs [35,36,37]. Besides that, since the beginning of the COVID-19 pandemic, several world leaders have claimed Azithromycin might be useful against COVID-19 [6,168].

Although contrasting data, some countries, such as Jordan and Croatia, reported increased prescription and distribution of Azithromycin to treat outpatients with COVID-19 in early 2021 and 2020, respectively [169,170]. In Brazil, however, although the widespread use of antibiotics increased from 2019–2020 to treat patients in the intensive care unit, no statistical significance was observed in Azithromycin use [171]. In contrast, data from the USA are conflicting.

A study showed a decreased prescription of antibiotics in early 2020 (January to May) compared to 2019, including a decrease in 33% of the Azithromycin prescriptions [172]. Another study, also performed in the USA, observed an increase in the prescription of Azithromycin between February and March 2020 compared to 2019, ranging from 1.3% to 8.7%. However, from March to April, the prescription rate decreased, ranging from (-)12% to (-)62.7% [158]. On the other hand, data from a single center in New York City showed an increase in the prescription of antibiotics to outpatients, ranging from 31.94 prescriptions per 1000 visits from March to May 2019 to 57.48 prescriptions per 1000 visits in 2020, which might be due to the increase of Azithromycin prescriptions, which increased from 5.88 prescriptions per 1000 visits to 7.16 prescription per 1000 visits [173].

Italy also has conflicting data, although a report showed an increased prescription of Azithromycin in 2022 to treat mainly the Omicron variant, leading to a shortage of this antibiotic [174]. Another study performed in the Emilia-Romagna region (Italy) observed a decrease in overall antibiotics consumption between March and May 2020 compared to 2019. Although a slight increase in the prescription of Azithromycin was observed in March 2020 (+24%), it started to decrease in April [(-)4%] and May [(-)48%] [175].

There are many conflicting data regarding the prescription of antibiotics in the COVID-19 pandemic, especially Azithromycin, a drug hypothesized to treat COVID-19. The lower prescription of antibiotics might have been influenced by the COVID-19 lockdown, which might have impacted bacterial infection dynamics. It might also have prevented individuals from seeking ambulatory help in a mild illness, thus preventing erroneous prescriptions [175]. On the other hand, the prescription of antibiotics, especially Azithromycin, to individuals with COVID-19 is problematic since this regime should only be instituted in individuals with documented bacterial infection; however, a low prevalence of co-infection/co-detection between SARS-CoV-2 and bacteria was reported [176,177,178], thus not supporting the routine use of Azithromycin in individuals with COVID-19. Unfortunately, a recent meta-analysis observed a discrepancy between the antibiotic prescription in individuals with COVID-19 with bacterial infection, being that among 30,000 individuals with COVID-19, only 8.6% also had a bacterial infection; however, 75% of them were prescribed antibiotics, including Azithromycin [179].

The excessive use of antibiotics, especially Azithromycin, might be responsible for changes in the human gut microbiota and enhance bacterial resistance, not only to macrolide but also to non-macrolide antibiotics [174,180,181]. A recent study in Mexico observed increased bacterial antibiotic resistance, including for other macrolides, such as Erythromycin and Clindamycin, in COVID-19 centers [182]. As aforementioned, the inadvertent use of Azithromycin should be appropriately addressed since it might contribute to bacterial resistance.

Another issue observed in the wide use of Azithromycin is its adverse effects. Although the most common adverse effects are gastrointestinal, like nausea, diarrhea, vomiting, and abdominal pain [183]; it can also cause more serious adverse effects as prolonged QTc intervals, even more, so when combined with HCQ, which is a usual drug of the COVID-kit in Brazil [104,107], which can predispose polymorphic ventricular tachycardia (*Torsades de Pointes*), a malignant arrhythmia which can cause death. Thus, the use of Azithromycin should not be done routinely to treat patients with COVID-19 to prevent bacterial resistance and QTc prolongation.

Finally, it is important to evaluate the publication of clinical trials for repurposed drugs during the COVID-19 pandemic because Scientific Production worldwide was affected by the pandemic, and it was associated with a high number of retracted papers including the clinical trials [119,184,185]. Also, in the future, it is important to perform other clinical trials and quality observational studies as those performed by the National Institute for Health and Care Research Global Health Unit on Global Surgery and COVIDSurg Collaborative (https://globalsurg.org/covidsurg/—accessed on 22 August 2024) to improve the world capacity to deal with conditions such as COVID-19 pandemic [186,187,188,189,190].

### 3.5. Limitations

There is a lot of confusion regarding what the anticipated benefit is likely to be because the literature used several outcomes, such as mortality, disease progression, need for ICU, and reduction in symptoms. Besides the presence of hospitalization, there are disparities in severity among the participants of the studies. Discrepancy among the studies for the use of monotherapy versus polypharmacy. Most of the articles included were heterogenous toward the “standard of care” treatment, which might make difficult comparisons. We only included three widely used drugs to treat COVID-19; we left some other drugs that might have impacted the pandemic management, such as Vitamin D, Oseltamivir, and Nitazoxanide. We did not perform a meta-analysis to assess the cumulative effect of these drugs. We also did not pre-register our systematic review on PROSPERO (International Prospective Register of Systematic Reviews). We only included phase III clinical trials that treated inpatients, this might have limited our analysis to only hospitalized patients.

## 4. Conclusions

COVID-19 was one of the deadliest pandemics in modern human history. Due to the potential health catastrophe caused by SARS-CoV-2, a global effort was made to evaluate treatments for COVID-19 to attenuate its impact on the human species. Unfortunately, several countries prematurely justified the emergency use of drugs that showed only in vitro effects against SARS-CoV-2, with a dearth of evidence supporting efficacy in humans. In this context, the purpose of this systematic review was to evaluate the mechanisms of several drugs proposed to treat COVID-19, including Ivermectin, CQ/HCQ, and Azithromycin, and to describe systematically phase III clinical trials that evaluated the efficacy of these drugs for treating patients with this respiratory disease. As the main finding, although Ivermectin, CQ/HCQ, and Azithromycin might have mechanistic effects against SARS-CoV-2 infection, most phase III clinical trials observed no treatment benefit in patients with COVID-19. The evidence doesn’t support the efficacy of these drugs in the treatment of COVID-19, based on the reviewed phase III trials. In this context, future robust studies with greater scientific rigor are needed to focus on new effective treatments.

## Figures and Tables

**Figure 1 biomedicines-12-02206-f001:**
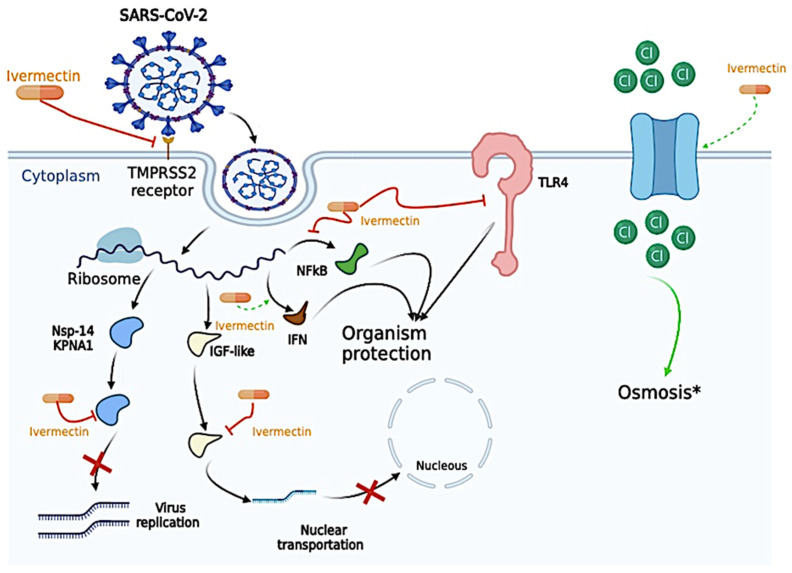
**Proposed antiviral mechanism of Ivermectin**. Ivermectin can disrupt the binding of essential proteins that allow cell entrance, such as Transmembrane Serine Protease 2 (TMPRSS2) and the Spike Protein. Ivermectin was also described to (i) bind to the alpha subunit of the insulin-like growth factor (IGF) superfamily and prevent the nuclear transportation of the severe acute respiratory syndrome coronavirus 2 (SARS-CoV-2); (ii) generate apoptosis and osmotic cell death by upregulating chloride channels since Ivermectin molecules behave as ionophores. In the same way, Ivermectin was able to bind to essential proteins for viral replication, such as nonstructural protein 1 (nsp-14) and Karyopherin-α1 (KPNA1), thus decreasing viral replication activity. Ivermectin also plays a vital role in several pro-inflammatory and anti-inflammatory cytokines, as inhibition of Toll-Like Receptors (TLRs), especially the TLR-4, blockade the nuclear factor kappa-light-chain-enhancer of activated B cells (NF-kB) transcriptional pathway, which might “protect” the host cell from the SARS-CoV-2 infection. IFN, interferon. *, Ivermectin is able to increase cell osmosis, and, in the figure, we exemplify its effect through the passage of Chloride (Cl). The figure was created in BioRender (BioRender.com).

**Figure 2 biomedicines-12-02206-f002:**
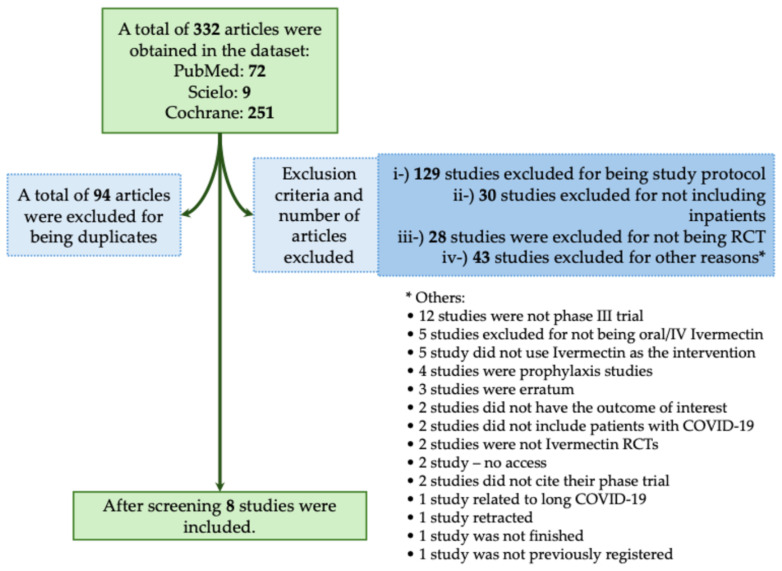
**Systematic review flowchart of clinical trials using Ivermectin during the coronavirus disease (COVID)-19 pandemic.** We included in our systematic review a total of eight studies (Okumuş et al., 2021; Shakhsi Niaee et al., 2021; Beltran Gonzalez et al., 2022; Heydari et al., 2022; Lim et al., 2022; Qadeer et al., 2022; Rezai et al., 2022; Baghbanian et al., 2023) [17,18,19,20,21,22,23,24]. The data search was performed on PubMed-Medline, Cochrane, and SciELO from COVID-19 pandemic onset to December 2023. The following search was performed: Ivermectin: (((Ivermectin)) AND ((COVID-19) OR (COVID-19 treatments) OR (COVID-19 pandemic) OR (SARS-CoV-2) OR (SARS-CoV-2 infection))) AND (Therapy/Narrow[filter]) AND (randomized controlled trial[pt] OR controlled clinical trial[pt] OR clinical trials as topic[mesh:noexp] OR trial[ti] OR random*[tiab] OR placebo*[tiab]). RCT, randomized controlled trial; IV, intravenous. *, The 45 studies that were excluded from different criteria were presented separately due to the low number of studies per criteria.

**Figure 3 biomedicines-12-02206-f003:**
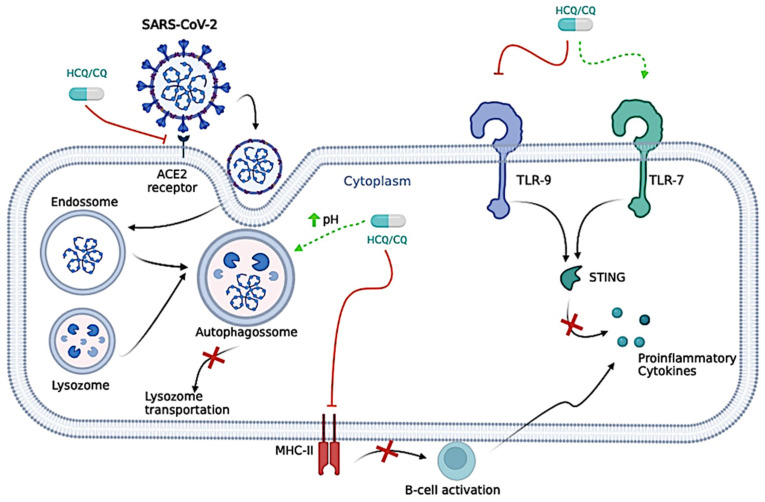
**Proposed antiviral mechanisms of Chloroquine/Hydroxychloroquine (CQ/HCQ).** CQ/HCQ is responsible for neutralizing the pH of the lysosome, which could prevent vital viral pathways, such as the S protein cleavages, and make it difficult for the virus to enter host cells. CQ/HCQ is also responsible for the inhibition of the lysosomes and autophagosomes, ultimately leading to the blockage of lysosome transportation. CQ/HCQ inhibits the movement of severe acute respiratory syndrome coronavirus 2 (SARS-CoV-2) from early endosomes to early lysosomes, thus further disrupting the release of viral genetic material. CQ/HCQ reduces Angiotensin-Converting Enzyme (ACE)-2 glycosylation, inhibiting this interaction and preventing the virus from binding and integrating into new cells. CQ/HCQ might also interfere in the cytokine storm caused by coronavirus disease (COVID)-19 through several pathways, such as the inhibition of the presentation of antigen by the antigen-presenting cells to T cells, declining the total of T cell activations, the blockage of Toll-Like Receptor (TLR)-9, activation of TLR-7, interfering in the Stimulator of Interferon Gene (*STING*) pathway—cyclic guanosine monophosphate–adenosine monophosphate (GMP-AMP) synthase, and ultimately decreasing the production of cytokines. The processing of antigens to peptides in Antigen Presenting Cells (APCs) is disrupted by HCQ, which, in turn, disrupts peptide presentation for major histocompatibility complex class II (MHC-II) cells. The disruption caused by HCQ also interferes with B-cell activation by CD4+ (cluster of differentiation 4) T-cells; thus, this diminishes their functions and cytokine production [Interleukin (IL)-1, IL-6, Interferon (INF)-gamma (INF-γ), TNF-alpha (TNF-α), and B-cell activating factor]. The figure was created in BioRender (BioRender.com).

**Figure 4 biomedicines-12-02206-f004:**
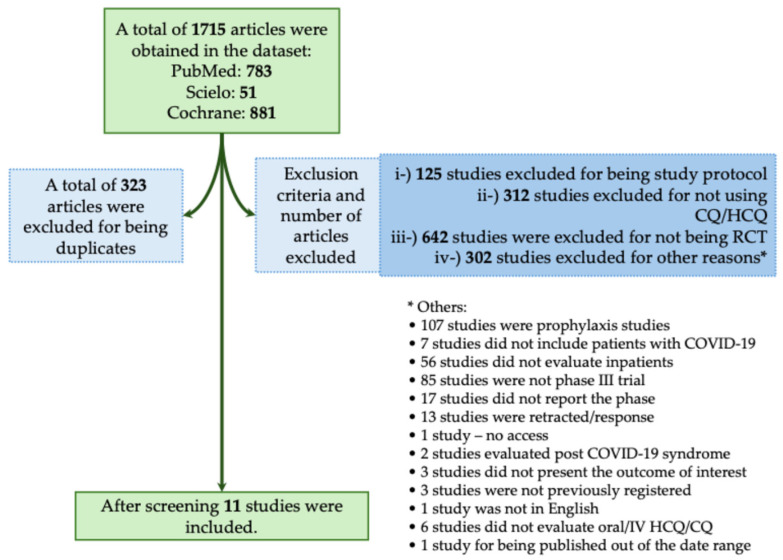
**Systematic review flowchart of clinical trials using Chloroquine/Hydroxychloroquine (CQ/HCQ) during the coronavirus disease (COVID)-19 pandemic.** We included in our systematic review a total of 11 studies according to the inclusion criteria (Horby et al., 2020; NCT04358081, 2020; Self et al., 2020; Ader et al., 2021; Arabi et al., 2021; Dubée et al., 2021; Hernandez-Cardenas et al., 2021; Pan et al., 2021; Réa-Neto et al., 2021; Ader and DisCoVeRy Study Group, 2022; Beltran Gonzalez et al., 2022) [19,25,26,27,28,29,30,31,32,33,34]. The data search was performed on PubMed-Medline, Cochrane, and SciELO from the COVID-19 pandemic onset to December 2023. The following search was performed: (((Chloroquine) OR (Hydroxychloroquine)) AND ((COVID-19) OR (COVID-19 treatments) OR (COVID-19 pandemic) OR (SARS-CoV-2) OR (SARS-CoV-2 infection))) AND (Therapy/Narrow[filter]) AND (randomized controlled trial[pt] OR controlled clinical trial[pt] OR clinical trials as topic[mesh:noexp] OR trial[ti] OR random*[tiab] OR placebo*[tiab]). RCT, randomized controlled trial; IV, intravenous. *, The 295 studies that were excluded from different criteria were presented separately due to the low number of studies per criteria.

**Figure 5 biomedicines-12-02206-f005:**
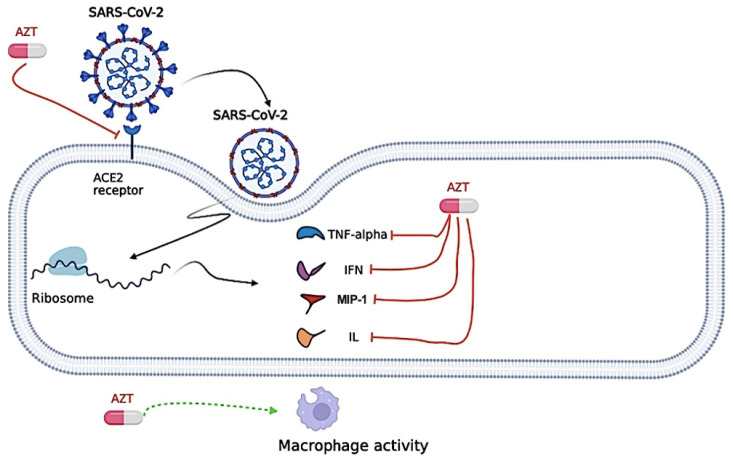
**Proposed antiviral mechanism of Azithromycin (AZT).** Azithromycin seems to negatively regulate the inflammatory cascade, attenuating the excessive production of cytokines [Tumor necrosis factor alpha (TNF-α), Interferon (INF), Interleukin (IL), and Macrophage Inflammatory Protein-1 (MIP-1)] in viral infections. Azithromycin can also influence phagocytic activity by modifying several functions, including chemotaxis, phagocytosis, oxidative burst, bacterial killing, and cytokine production. Azithromycin could interfere with the binding between the severe acute respiratory syndrome coronavirus 2 (SARS-CoV-2) Spike protein and the Angiotensin Converting Enzyme (ACE)-2 receptor protein, preventing the virus from entering the cell. The figure was created in BioRender (BioRender.com).

**Figure 6 biomedicines-12-02206-f006:**
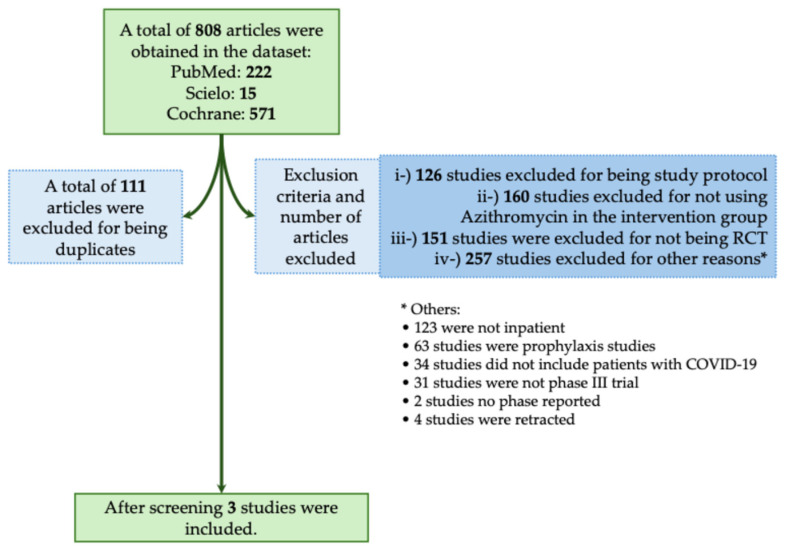
**Systematic review flowchart of clinical trials using Azithromycin during the coronavirus disease (COVID)-19 pandemic**. We included in our systematic review a total of three studies (Cavalcanti et al., 2020; Furtado et al., 2020; RECOVERY Collaborative Group, 2021) [35,36,37]. The data search was performed on PubMed-Medline, Cochrane, and SciELO from the COVID-19 pandemic onset to December 2023. The following search was performed: (((Azithromycin) OR (Antibiotics) OR (Macrolides)) AND ((COVID-19) OR (COVID-19 treatments) OR (COVID-19 pandemic) OR (SARS-CoV-2) OR (SARS-CoV-2 infection))) AND (Therapy/Narrow[filter]) AND (randomized controlled trial[pt] OR controlled clinical trial[pt] OR clinical trials as topic[mesh:noexp] OR trial[ti] OR random*[tiab] OR placebo*[tiab]). RCT, randomized controlled trial. *, The 257 studies that were excluded from different criteria were presented separately due to the low number of studies per criteria.

**Table 1 biomedicines-12-02206-t001:** Description of the phase III randomized clinical trials that assessed Ivermectin as a treatment against coronavirus disease (COVID)-19.

Authors	Sample	Randomized Participants	Center	Groups	Blindness	Dosage/Duration	Primary Outcome	Results	Conclusions
Okumuş et al. (2021) [17]	Hospitalized severe COVID-19 individuals with pneumonia.	A total of 66 individuals were recruited.	Single center (Afyonkarahisar Health Science University).	Two arms: (i) Ivermectin plus reference treatment; (ii) Standard of treatment.	Single-blind.	Two arms: (i) 200 mcg/kg of Ivermectin for five days plus reference treatment prepared by the Turkish Ministry of Health. (ii) Standard of care prepared by the Turkish Ministry of Health.	Clinical response on the fifth day (Extubation in mechanically ventilated individuals, respiratory rate <26, SpO_2_ level in room air >90%, PaO_2_/FiO_2_ >300 in individuals receiving oxygen, presence of at least two of the 2-point reduction criteria in SOFA) and drug side effects.	At the end of the 5-day follow-up: 22/30 (73.3%) of the study population improved compared to 16/30 (53.3%) in the control group. Regarding mortality, 6 (20%) of the individuals died compared to 9 (30%) in the control group. The differences observed were not statistically significant. When the mean SOFA scores before treatment and at the end of the follow-up period were compared, a significant decrease was found only in the study group. However, when the SOFA scores of both groups were compared at the end of the follow-up period, no significant difference was described. At the end of the study, 16 (57.1%) individuals in the study group and 8 (26.7%) in the control group were investigated by PCR test for SARS-CoV-2. Of these individuals, 14 (87.5%) individuals in the study group and 3 (37.5%) individuals in the control group were found to become negative. A significant increase was observed in PaO_2_/FiO_2_ ratios in the study group compared to the initial values. The increase in PaO_2_/FiO_2_ ratios continued in both groups during the follow-up period and at the end of the follow-up period, the increase in the study group according to the baseline values was again found to be significant. C-reactive protein, ferritin, and D-dimer markers presented better values in the study group than in the control group at the end of the follow-up period	Ivermectin did not improve statistically significant clinical response or mortality.
Shakhsi Niaee et al. (2021) [18]	Hospitalized individuals with mild COVID-19 confirmed by RT-PCR or chest images.	A total of 180 individuals were recruited.	Multicenter (Qazvin and Khuzestan).	Six arms (i) Hydroxychloroquine; (ii) Placebo + Hydroxychloroquine; (iii) Single dosage of Ivermectin; (iv) Interval Ivermectin with low dosages; (v) Single higher dosage of Ivermectin; (vi) Interval Ivermectin with higher dosages.	Double-blind.	(i) Hydroxychloroquine 200-mg twice per day. (ii) Placebo plus Hydroxychloroquine 200-mg twice per day. (iii) Single dose of Ivermectin 200 mcg/kg. (iv) Three low interval dosages of Ivermectin (200, 200, and 200 mcg/kg). (v) Single dose of Ivermectin (400 mcg/kg). (vi) Three high-interval doses of Ivermectin (400, 200, and 200 mcg/kg).	All-cause mortality or clinical recovery.	The duration of low SpO_2_ and hospital stay was lower for the groups of individuals that received a single dose of Ivermectin (200 mcg/kg or 400 mcg/kg) compared to those that received Hydroxychloroquine 200-mg twice per day or placebo plus Hydroxychloroquine 200-mg twice per day. Also, the participants who received Ivermectin were less prone to die. In brief, the occurred for 5 (16.7%), 6 (20.0%), 0 (0.0%), 3 (10.0%), 0 (0.0%), and 1 (3.3%), respectively for the individuals that received Hydroxychloroquine only, Hydroxychloroquine plus placebo, single dose of Ivermectin (200 mcg/kg), three low interval dosages of Ivermectin, single dose of Ivermectin (400 mcg/kg), and three high interval doses of Ivermectin.	Ivermectin as an adjunct may reduce mortality rate, time of low SpO_2_, and duration of hospitalization.
Beltran Gonzalez et al. (2022) [19]	Hospitalized individuals.	A total of 106 participants were recruited.	Single center (Mexico).	Three arms: (i) Hydroxychloroquine; (ii) Ivermectin; (iii) Placebo.	Double-blind.	(i) Hydroxychloroquine, 400-mg every 12-h on the first day and, subsequently, 200-mg every 12-h for four days. (ii) Ivermectin, 12- or 18-mg, according to patient weight. (iii) Placebo.	Length of hospital stay, death, and respiratory deterioration.	No difference in hospital stay was observed between the treatment groups, nor in respiratory deterioration or chance of death.	In hospitalized individuals with COVID-19 Ivermectin did not influence hospital length of stay, death, or respiratory deterioration.
Heydari et al. (2022) [20]	Hospitalized individuals with COVID-19 confirmed by RT-PCR or chest image.	A total of 107 individuals were recruited.	Single center.	Three arms: (i) Ivermectin plus standard of treatment; (ii) Metronidazole plus standard of treatment; (iii) Standard treatment.	Triple-blind.	(i) Oral Ivermectin 200 mcg/kg. (ii) Metronidazole 8 mg/kg 6/6-h for five days. (iii) Standard of treatment.	(i) Vital signs (body temperature, respiratory rate, heart rate, systolic blood pressure, diastolic blood pressure, and SpO_2_); (ii) Biomedical parameters such as the levels of lymphocytes, neutrophils, platelets, and white blood cells; (iii) Length of hospital stay and death.	The mortality rate in Ivermectin was lower compared to the other groups (4.5% versus 15.8% versus 11.8%), but not statistically significant. After five days, the mean difference in lymphocyte and neutrophil count was significantly different between groups. The other characteristics were not significant.	Ivermectin did not improve patients’ recovery compared to standard of care alone.
Lim et al. (2022) [21]	Hospitalized in a quarantine hospital.	A total of 490 participants were recruited.	Multi-center (Malasya).	Two arms: (i) Ivermectin; (ii) Standard of care.	Open-label.	(i) Four dosages of Ivermectin were used varying from 6- to 12-mg for five days. (ii) Standard of care.	The proportion of individuals who progressed to severe disease, defined as the hypoxic stage requiring supplemental oxygen to maintain SpO_2_ of 95% or higher.	Ivermectin did not prevent the progress to severe disease compared to standard of care [RR (95%CI) = 1.25 (0.87 to 1.80)]. Regarding the secondary outcomes, Ivermectin did not improve the need for mechanical ventilation support [RR (95%CI) = 0.41 (0.13 to 1.30)], intensive care [RR (95%CI) = 0.78 (0.27 to 2.20)] and 28-day in-hospital mortality [RR (95%CI) = 0.31 (0.09 to 1.11)].	Ivermectin, when compared to standard care, did not reduce disease severity, need for mechanical ventilation, intensive care, or 28-day in-hospital mortality.
Qadeer et al. (2022) [22]	COVID-19-confirmed individuals treated in a COVID-19 treatment center.	A total of 210 individuals were recruited.	Single center (Pakistan).	Two arms: (i) Ivermectin + standard of treatment; (ii) Placebo according to international guidelines.	Not informed.	(i) Ivermectin two tablets of 6-mg once daily for five days plus standard of care. (ii) Standard of care.	Time of viral clearance measured by COVID-19 RT-PCR on days seven, 14, and 21.	A total of 21 (20%) individuals in the Ivermectin group had negative RT-PCR on day seven, while in the placebo group, all 105 individuals still tested positive for COVID-19. On day 10, 70 (66.7%) individuals in the Ivermectin group had negative RT-PCR for COVID-19 versus 21 (20%) in the placebo group. On day 14, all of the individuals in the Ivermectin group tested negative versus 70 (66.7%) individuals in the placebo group.	Ivermectin may improve viral clearance compared to placebo.
Rezai et al. (2022) [23]	Non-hospitalized and hospitalized adults with a positive RT-PCR test for COVID-19.	A total of 609 inpatients and 549 outpatients were recruited. ^+^	Two multicenter studies were conducted for inpatients (seven hospitals in six cities) and outpatients.	Two arms: (i) Oral Ivermectin for three days plus standard of care; (ii) Placebo plus standard of care.	Double-blind.	- 0.4 mg/kg of body weight per day for three days. - In the control group, placebo tablets were used for three days.	Time to resolution of symptoms, recovery including complete recovery (resolving main complaints on the seventh day) and relative recovery (remaining main complaints on the seventh day).	Complete recovery was higher in the Ivermectin group (37%) compared to the placebo group (28%) [RR (95%CI) = 1.32 (1.04 to 1.66)]. Length of hospital stay was significantly longer in the Ivermectin group (7.98 ± 4.4) compared to the control group (7.16 ± 3.2). There was no difference in need for intensive care unit [RR (95%CI) = 0.84 (0.52 to 1.36)], need for invasive mechanical ventilation [RR (95%CI) = 0.50 (0.24 to 1.07]), need for noninvasive mechanical ventilation [RR (95%CI) = 0.93 (0.86 to 1.00) and chance of death [RR (95%CI) = 0.69 (0.35 to 1.39).	Ivermectin, when compared to a placebo, did not improve the need for an intensive care unit, the need for noninvasive or invasive mechanical ventilation, and clinical involvement in hospitalized COVID-19 individuals.
Baghbanian et al. (2023) [24]	Intubated COVID-19 individuals.	A total of 60 individuals were recruited.	Single center.	Two arms: (i) Placebo; (ii) Ivermectin.	Double-blind.	(i) Ivermectin 6-mg twice a day on the first day and from the second to the fifth day 30-mg twice a day. (ii) Placebo.	Mortality.	There was no difference in mortality between the two groups. Regarding secondary outcomes, the heart rate showed a decrease in the Ivermectin group compared to placebo group on day two and SpO_2_ showed an improvement in the Ivermectin group compared to the placebo group on the fifth day and sixth day, all other vital signs were not significant.	Ivermectin did not improve mortality in hospitalized individuals with COVID-19.

%, percentage; 95%CI: 95% Confidence Interval; FiO_2_, fraction of inspiratory oxygen concentration; h, hour; kg, kilogram; mg, milligram; mcg, microgram; PaO_2_, pressure of oxygen in arterial blood; RR, relative risk; RT-PCR, real-time polymerase chain reaction; SARS-CoV-2, severe acute respiratory syndrome coronavirus 2; SOFA, Sequential Organ Failure Assessment; SpO_2_, peripheral arterial oxygen saturation. ^+^ Since this paper evaluated both in- and outpatients, we only analyzed inpatients.

**Table 2 biomedicines-12-02206-t002:** Risk of bias summary for randomized studies (RoB 2) that used Ivermectin.

Study	Bias from the Randomization Process	Bias Due to Deviations from Intended Interventions	Bias Due to Missing Outcome Data	Bias in the Measurement of the Outcomes	Bias in Selection of the Reported Result	Overall Risk of Bias
Okumuş et al. (2021) [17]	High	Low	Low	High	High	High
Shakhsi et al. (2021) [18]	Some concerns	Low	Low	High	High	High
Beltran Gonzalez et al. (2022) [19]	Low	Low	Low	Low	Low	Low
Heydari et al. (2022) [20]	Low	Low	Low	Low	Some concerns	Some concerns
Lim et al. (2022) [21]	Low	Low	Low	Low	Low	Low
Quadeer et al. (2021) [22]	Some concerns	Some concerns	Low	High	Low	High
Rezai et al. (2022) [23]	Low	Low	Low	Low	Low	Low
Baghbanian et al. (2023) [24]	Low	Some concerns	Low	Low	Low	Some concerns

**Table 3 biomedicines-12-02206-t003:** Description of the phase III randomized clinical trials that assessed Chloroquine/Hydroxychloroquine (CQ/HCQ) as a treatment against coronavirus disease (COVID-19).

Author	Sample	Randomized Participants	Center	Groups	Blindness	Dosage/Duration	Primary Outcome	Results	Conclusions
Beltran Gonzalez et al. (2022) [19]	Hospitalized COVID-19 individuals.	A total of 106 participants were recruited.	Single Center (Mexico).	Three arms: (i) HCQ; (ii) Ivermectin; (iii) Placebo.	Double-blind.	(i) HCQ, 400-mg every 12-h on the first day and, subsequently, 200-mg every 12-h for four days. (ii) Ivermectin, 12-mg or 18-mg, according to patient weight. (iii) Placebo.	The study evaluated the length of hospital stay, the number of deaths, and the presence of respiratory deterioration. Safety outcomes: Tolerance and adverse effects.	No significant difference in hospital stay was observed between the treatment group (group i: seven versus group iii: five days); nor in respiratory deterioration (group i: 6 (18;1%) versus group iii: 9 (24.3%)) nor death [group i: 2 (6%) versus group iii: 6 (16.2%)].	In hospitalized individuals with COVID-19 HCQ did not have an effect on hospital length of stay, death, or respiratory deterioration.
RECOVERY Collaborative Group et al. (2020) [25]	Hospitalized participants with clinically suspected or laboratory-confirmed COVID-19.	A total of 4716 participants were recruited.	Platform trial (176 hospitals in the United Kingdom).	(i) 1561 participants received HCQ; (ii) 3155 participants received usual care. The remainder of the participants were randomly assigned to one of the other treatment groups.	Not blind.	- HCQ sulfate (200-mg tablet) four tablets (800-mg) at baseline and 6-h, followed by two tablets (400-mg) starting at 12-h after the initial dose and then every 12-h for the next nine days or until discharge, whichever occurred earlier. - Usual care.	All-cause mortality on 28 days after randomization.	Death on 28 days occurred in 421 of 1561 (27.0%) participants in the HCQ group and 790 of 3155 (25.0%) participants in the usual-care group. The difference in the death rates was not statistically significant.	The HCQ group did not have a lower incidence of death 28 days post-randomization compared to those who received usual care.
Self et al. (2020) [26]	Hospitalized participants with positive SARS-CoV-2 RT-PCR.	A total of 479 participants were recruited.	Multicenter (34 hospitals in the USA).	(i) 242 participants received HCQ; (ii) 237 participants received a placebo.	Not informed.	- 400-mg HCQ twice a day for the first two doses, then 200-mg of HCQ twice a day for eight doses. - Placebo.	Clinical status at day 14 (the scale of the COVID-19 outcome was used).	The median interquartile score of clinical status at day 14: six (HCQ) versus six (placebo) [OR (95%CI) = 1.02 (0.73 to 1.42)].	HCQ did not improve clinical status on day 14.
Ader et al. (2021) [27]	Hospitalized participants with positive SARS-CoV-2 RT-PCR and pulmonary crackles or SpO_2_ ≤ 94% or who required supplemental oxygen.	A total of 603 participants were recruited.	Multicenter (Academic or non-academic hospitals throughout Europe).	(i) 152 received standard of care; (ii) 150 received standard of care plus Lopinavir/Ritonavir; (iii) 150 received standard of care plus Lopinavir/Ritonavir plus Interferon-β-1a; (iv) 151 received standard of care plus HCQ.	Open-label.	- 400-mg Lopinavir and 100-mg Ritonavir twice a day for 14 days. - 44-mcg Interferon-β-1a on days one, three, and six. - 400-mg HCQ twice on day one and 400-mg daily for nine days.	Clinical status on day 15 was measured on the seven-point ordinal scale of the WHO.	Death at the 15-day: 7 (7%) versus 5 (5%), for control and HCQ, respectively [OR (95%CI) = 0.93 (0.62–1.41)].	HCQ did not improve clinical status on day 15 compared to the control.
Arabi et al. (2021) [28]	Individuals with confirmed or suspected COVID-19 and who were also receiving respiratory or cardiovascular organ failure support in the intensive care unit.	A total of 726 participants were recruited.	Multicenter (Canada, USA, France, Germany, Ireland, Netherlands, Portugal, United Kingdom, Saudi Arabi, Australia, and New Zealand).	(i) 268 participants were assigned to Lopinavir/Ritonavir (249 included in the final analysis); (ii) 52 participants were assigned to HCQ (49 included in the final analysis); (iii) 29 participants were assigned to Lopinavir/Ritonavir plus HCQ (26 included in the final analysis); (iv) 377 participants were assigned to receive no antiviral drug (353 included in the final analysis).	Blind.	- 400-mg Lopinavir and 100-mg Ritonavir every 12-h for five to 14 days. - Two doses of 800-mg of HCQ 6-h apart, followed 6-h later by 400-mg 12 hourly for 12 doses.	The ordinal scale of the number of respiratory and cardiovascular organ support-free days and hospital mortality.	The median organ support-free days among participants in Lopinavir-Ritonavir, HCQ, and combination therapy groups were 4 (−1 to 15) [OR (95%CI) = 0.73 (0.55 to 0.99)], 0 (−1 to 9) [OR (95%CI) = 0.57 (0.35 to 0.83)], and −1 (−1 to 7) [OR (95%CI) = 0.41 (0.24 to 0.72)], respectively, compared to 6 (−1 to 16) days in the control group. In-hospital mortality among participants in Lopinavir-Ritonavir, HCQ, and combination therapy was 88/249 (35.3%) [OR (95%CI) = 0.65 (0.45 to 0.95)], 17/49 (34.7%) [OR (95%CI) = 0.56 (0.36 to 0.89)], and 13/26 (50%) [OR (95%CI) = 0.36 (0.17 to 0.73)], respectively, compared to 106/353 (30%) in the control group.	In critically ill individuals infected with SARS-CoV-2, treatment with Lopinavir-Ritonavir, HCQ, or combination therapy resulted in worse outcomes compared to no COVID-19 antiviral therapy.
Dubée et al. (2021) [29]	Men and non-pregnant women aged +18 years with COVID-19 were confirmed by positive SARS-CoV-2 RT-PCR or chest computed tomography scan with typical features of COVID-19. Participants: (a) need for supplemental oxygen; (ii) age ≥75 years; and (iii) age between 60 and 74 years and presence of at least one co-morbidity.	A total of 250 participants were recruited.	Multicenter (France and Monaco).	(i) 110 participants received HCQ; (ii) 116 participants received a placebo.	Double-blind.	- 800-mg (two tablets, twice daily) on the first day and one 200-mg tablet twice daily for the following eight days (four grams total). - Placebo.	Composite endpoint: Death and the need for invasive mechanical ventilation within 14 days following randomization.	The primary endpoint occurred in 9 (7.3%) versus 8 (6.5%) in HCQ and control groups, respectively [RR (95%CI) = 1.23 (0.43 to 3.55)].	In order of the study’s premature discontinuation and lower-than-expected primary outcomes, no significant conclusion can be drawn on the efficacy of HCQ.
Hernandez-Cardenas et al. (2021) [30]	Age +18 years with COVID-19 confirmed by SARS-CoV-2 RT-PCR, symptoms onset <14 days, with lung injury requiring hospitalization with or without mechanical ventilation.	A total of 214 participants were recruited.	Multicenter (Mexico).	(i) 106 participants received HCQ; (ii) 108 participants received placebo.	Double-blind.	- HCQ orally or by nasogastric tube—200-mg 12/12-h for 10 days. - Placebo.	The 30-day mortality rate after randomization.	The 30-day mortality rate was 38% and 41% in the HCQ and placebo groups, respectively with a Hazard ratio of 0.89 (95%CI = 0.58 to 1.38).	No benefit or significant harm using HCQ can be demonstrated in this placebo-controlled randomized trial.
Pan et al. (2021) [31]	It was included participants ≥18 years, hospitalized with COVID-19, not known to have received any trial drug, not expected to be transferred elsewhere within 72-h, and no contraindication to any trial drug.	A total of 1863 participants were recruited.	Multicenter (405 hospitals in 30 countries).	(i) 954 participants received HCQ; (ii) 909 participants did not receive HCQ.	Not blind.	- HCQ Sulfate: four tablets (800-mg) at hour zero, four tablets at hour six and starting at hour 12, two tablets (400-mg) twice daily for 10 days. - Placebo.	In-hospital mortality in the four pairwise comparisons of each trial drug and its control. Regardless of whether it occurred before or after day 28.	No trial drug had a significant definite effect on mortality. The death occurred in 104 of 947 participants receiving HCQ and in 84 of 906 participants receiving its control.	HCQ regimen had little or no effect on hospitalized participants with COVID-19.
Réa-Neto et al. (2021) [32]	Individuals who were admitted to the intensive care unit or acute care room with flu symptoms and dyspnea or need for supplemental oxygen or SpO_2_ ≤94 on room air or computerized tomography scan compatible with COVID-19 or need for mechanical ventilation and a confirmed diagnosis of COVID-19.	A total of 142 individuals were randomized but only 105 subjects were evaluated in the modified intention to treat.	Multicenter (Brazil).	Two groups: (i) CQ/HCQ plus standard of care; (ii) Placebo plus standard of care.	Open Label.	(i) CQ 450-mg twice a day on day one followed by 450-mg once a day from day 2–5 OR HCQ 400-mg twice a day followed by 400-mg once daily from day 2–5 plus standard of care. (ii) Placebo plus standard of care.	Clinical status on day 14 with a 9-point ordinal scale.	On the 14th day, the odds of having an unfavorable clinical outcome were higher in the CQ/HCQ group, even after controlling for confounding factors [OR (95%CI) = 2.45 (1.17 to 4.93)]. On the 28th day, individuals in the CQ/HCQ also presented worse clinical outcomes [OR (95%CI) = 2.47 (1.15 to 5.30)]. The mortality rate on the 28th day was not different between groups [(RR (95%CI) = 1.57 (0.79 to 3.13)].	CQ/HCQ appears to be associated with worse clinical outcomes in severe COVID-19 individuals.
Ader and DisCoVeRy Study Group (2022) [33]	COVID-19 inpatients requiring oxygen and/or ventilatory support.	A total of 603 participants were recruited.	Multicenter (30 sites in France and two in Luxembourg).	(i) standard of care; (ii) standard of care plus Lopinavir/Ritonavir; (iii) standard of care plus Lopinavir/Ritonavir plus IFN-β-1a; (iv) standard of care plus HCQ.	Not blind.	- Standard of care plus Lopinavir/Ritonavir (400-mg Lopinavir and 100-mg Ritonavir orally twice a day for 14 days). - Standard of care plus Lopinavir/Ritonavir plus Interferon-β-1a (44-mg subcutaneous Interferon -β-1a on days one, three, and six). - Standard of care plus HCQ (400-mg orally, twice on day one as a loading dose followed by 400-mg once daily for nine days).	Clinical status at day 15 as measured on the seven-point ordinal scale of the WHO Master Protocol (v3.0, 3rd March 2020).	Adjusted OR (aOR) (95%CI) for clinical improvement were not in favor of investigational treatments: Lopinavir/Ritonavir versus control [aOR (95%CI) = 0.83 (0.55 to 1.26)]; Lopinavir/Ritonavir plus Interferon-β-1a versus control [aOR (95%CI) = 0.69 (0.45 to 1.04)]; HCQ versus control [aOR (95%CI) = 0.93 (0.62 to 1.41)]. The occurrence of serious adverse events was higher in participants allocated to the Lopinavir/Ritonavir-containing arms.	In individuals admitted to hospital with COVID-19, Lopinavir/Ritonavir, Lopinavir/Ritonavir plus Interferon-β-1a and HCQ were not associated with clinical improvement at days 15 and 29, nor in a reduction in viral shedding, and generated more severe adverse side-effects in Lopinavir/Ritonavir-containing arms.
NCT04358081 (2020) [34] *	Hospitalized confirmed COVID-19 individuals.	A total of 20 ^§^ individuals were recruited.	Multicenter (USA).	Three groups: (i) HCQ plus placebo; (ii) HCQ plus azithromycin; (iii) Placebo.	Double-Blind.	(i) HCQ 600-mg once a day followed by 200-mg three times a day plus Azithromycin placebo. (ii) HCQ 600-mg once a day followed by 200-mg three times a day plus Azithromycin 500-mg followed by 250-mg once a day from days 2–5. (iii) HCQ and Azithromycin placebo.	Clinical response by day 15 was defined as discharged alive or no need for mechanical ventilation or no need for supplementary oxygen therapy.	On the 15th day, 7/7 (100%) individuals achieved clinical improvement in both groups i and ii, whereas in the group iii, 4/5 (80%) achieved clinical improvement.	All the individuals treated with HCQ with or without Azithromycin presented clinical improvement on the 15th day, compared to placebo.

95%CI, 95% confidence interval; %, percentage; aOR, adjusted odds ratio; h, hour; mcg, microgram; mg, milligram; ORs, odds ratio; RT-PCR, real-time polymerase chain reaction; SARS-CoV-2, severe acute respiratory syndrome coronavirus 2; SpO_2_, peripheral arterial oxygen saturation; USA, United States of America. ^§^ One of the patients was mis-randomized, so only 19 participants were evaluated. *, The study was obtained from the following website: https://clinicaltrials.gov/show/NCT04358081 (accessed on 22 August 2024).

**Table 4 biomedicines-12-02206-t004:** Risk of bias summary for randomized studies (RoB 2) that used CQ/HCQ.

Study	Bias from the Randomization Process	Bias due to Deviations from Intended Interventions	Bias Due to Missing Outcome Data	Bias in the Measurement of the Outcomes	Bias in Selection of the Reported Result	Overall Risk of Bias
Beltran González et al. (2022) [19]	Low	Low	Low	Low	Low	Low
Recovery (2020) [25]	Low	Low	Low	Low	Low	Low
Self et al. (2020) [26]	Low	Low	Low	Low	Low	Low
Ader et al. (2021) [27]	Low	Low	Low	Low	Low	Some concerns
Arabi et al., 2021 [28]	Some concerns	Low	Low	Low	Low	Low
Dubee et al. (2021) [29]	Low	Low	Low	Low	Low	Low
Hernandez Cardenas et al. (2021) [30]	Low	Low	High	Low	Low	High
Pan et al. (2021) [31]	Low	Low	Low	Low	Low	Low
Réa-Neto et al. (2021) [32]	Low	Low	Low	Low	Low	Low
Ader and DisCoVeRy Study Group, 2022 [33]	Low	Low	Low	Low	Low	Low
NCT04358081 (2020) [34] *	Some concerns	Low	High	Some concerns	Low	High

*, The study was obtained from the following website: https://clinicaltrials.gov/show/NCT04358081 (accessed on 22 August 2024).

**Table 5 biomedicines-12-02206-t005:** Description of the phase III randomized clinical trials which assessed Azithromycin as a treatment against coronavirus disease (COVID)-19.

Author	Sample	Randomized Participants	Center	Groups	Blindness	Dosage/Duration	Primary Outcome	Results	Conclusions
Cavalcanti et al. (2020) [35]	Hospitalized individuals with suspected or confirmed COVID-19 who were receiving either no supplemental oxygen or a maximum of 4 L/min of oxygen.	A total of 667 individuals were randomized and 504 had confirmed COVID-19.	Multicenter (Brazil).	Three groups: (i) Standard of care; (ii) Standard of care plus Hydroxychloroquine; (iii) Standard of care plus Azithromycin plus Hydroxychloroquine.	Open-label.	(i) Standard of care; (ii) Standard of care plus Hydroxychloroquine 400-mg twice daily for seven days. (iii) Standard of care plus Hydroxychloroquine 400-mg twice daily plus 500-mg once daily for seven days.	Clinical status at 15 days was assessed with the seven-level ordinal scale.	In the modified intention to treat (that is, only COVID-19 individuals), Hydroxychloroquine plus Azithromycin did not improve the clinical score at day 15 [OR (95%CI) = 0.99 (0.57 to 1.73).	The use of Hydroxychloroquine plus Azithromycin did not improve the clinical score in patients with COVID-19.
Furtado et al. (2020) [36]	Hospitalized participants with confirmed RT-PCR or suspected COVID-19 with at least one of the following characteristics: use of invasive mechanical ventilation OR noninvasive mechanical ventilation OR noninvasive positive pressure ventilation OR oxygen supplementation of more than 4 L/mL flow.	A total of 447 participants were recruited.	Multicenter (Brazil).	Two groups: (i) 237 participants received Azithromycin plus standard of care; (ii) 210 participants received standard of care.	Open-label.	- 500-mg Azithromycin once a day for 10 days. - Standard of care.	Clinical status at 15 days.	Azithromycin plus standard of care versus standard of care did not influence the clinical status at 15 days [OR (95%CI) = 1.36 (0.94 to 1.97)].	Adding Azithromycin to a standard of care did not result in clinical improvement in hospitalized COVID-19 participants.
RECOVERY Collaborative Group (2021) [37]	Hospitalized participants with confirmed RT-PCR or suspected COVID-19.	A total of 7763 participants were recruited.	Multicenter (176 hospitals in the UK).	Two groups: (i) 2582 participants received Azithromycin; (ii) 5181 participants received usual care.	Open-label.	- 500-mg Azithromycin once a day for 10 days. - Standard of care.	28-day-all-cause-mortality.	28-day-all-cause-mortality—N of people who died (%): 561 (22%) versus 1162 (22%) [Rate ratio (95%CI) = 0.97 (0.86 to 1.07)].	The results do not show Azithromycin is an effective treatment for hospitalized individuals with COVID-19.

95%CI, 95% confidence interval; %, percentage; L, liter; OR, odds ratio; mL, milliliters; mg, milligram; N, number of individuals; RT-PCR, real-time polymerase chain reaction; SARS-CoV-2, severe acute respiratory syndrome coronavirus 2; UK, United Kingdom.

**Table 6 biomedicines-12-02206-t006:** Risk of bias summary for randomized studies (RoB 2) that used Azithromycin.

Study	Bias from the Randomization Process	Bias Due to Deviations from Intended Interventions	Bias Due to Missing Outcome Data	Bias in the Measurement of the Outcomes	Bias in Selection of the Reported Result	Overall Risk of Bias
Cavalcanti et al. (2020) [35]	Low	Some concerns	Low	Low	Low	Some concerns
Furtado et al. (2020) [36]	Low	Some concerns	Low	Low	Low	Some concerns
Recovery (2021) [37]	Low	Low	Low	Low	Low	Low

## Data Availability

The original contributions presented in the study are included in the article, further inquiries can be directed to the corresponding author.

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
