# Peer review of "Efficacy of Ivermectin, Chloroquine/Hydroxychloroquine, and Azithromycin in Managing COVID-19: A Systematic Review of Phase III Clinical Trials"

_biomedicines, 2024, doi:10.3390/biomedicines12102206_

Round 1
Reviewer 1 Report
Comments and Suggestions for Authors
Dear authors,
Thank you for giving me the opportunity to review your work. as suggestions, I'll focus on the following aspects:
1. Title and Abstract:
- Title: Consider simplifying and making the title more concise. For example, "Efficacy of Ivermectin, Chloroquine/Hydroxychloroquine, and Azithromycin in Managing COVID-19: A Systematic Review of Phase III Clinical Trials."
- Abstract:
- Clarify the purpose of the review. Emphasize the aim of evaluating the clinical efficacy of these drugs rather than just describing mechanisms.
- Simplify language to enhance readability. For instance, rephrase "highlighting the importance of clinical trials" to "underscoring the need for robust clinical trials."
2. Introduction:
- Add more context about the global impact of COVID-19 and why these specific drugs were considered potential treatments.
- Clarify the rationale for including these three drugs. Mention briefly why other repurposed drugs (like Remdesivir) were not included.
3. Materials and Methods:
- Provide more details on the search strategy. For instance, mention the number of studies identified at each stage of the search.
- Explain the exclusion criteria more clearly, particularly why some studies were excluded.
4. Results:
- Ivermectin Section:
- Improve clarity by summarizing the results before delving into individual studies. For example, "The majority of studies found no significant benefit of Ivermectin in reducing mortality or severity in COVID-19 patients."
- In tables, ensure consistency in reporting units (e.g., mg vs. μg/kg).
- HCQ/CQ Section:
- Provide a similar summary at the beginning of this section. It helps the reader quickly grasp the overall findings.
- Consider a table that compares the outcomes of all three drugs side-by-side for a quick visual reference.
5. Discussion:
- Strengthen the discussion by comparing your findings with existing literature, particularly systematic reviews and meta-analyses.
- Highlight the limitations of the studies reviewed, including potential biases and the quality of evidence.
- Address the broader implications for clinical practice and policy, especially in light of the mixed evidence regarding these treatments.
6. Conclusion:
- Simplify and focus on the key takeaways. Reiterate that the evidence does not support the efficacy of these drugs in treating COVID-19, based on the reviewed phase III trials.
- Suggest future research directions, such as focusing on newer treatments or more rigorous trials.
7. Figures and Tables:
- Ensure all figures and tables are clearly labeled and easy to interpret.
- Simplify complex diagrams and use color-coding or labeling to clarify mechanisms of action.
Author Response
Comments and Suggestions for Authors
Dear authors,
Thank you for giving me the opportunity to review your work. As suggestions, I'll focus on the following aspects:
- Title and Abstract:
- Title: Consider simplifying and making the title more concise. For example, "Efficacy of Ivermectin, Chloroquine/Hydroxychloroquine, and Azithromycin in Managing COVID-19: A Systematic Review of Phase III Clinical Trials."
Reply: Dear reviewer, the title was corrected as recommended. The authors thank you for your important contribution to our study.
- Abstract:
- Clarify the purpose of the review. Emphasize the aim of evaluating the clinical efficacy of these drugs rather than just describing mechanisms.
Reply: Dear reviewer, we added an excerpt in the abstract section to clarify the purpose of the review.
- Simplify language to enhance readability. For instance, rephrase "highlighting the importance of clinical trials" to "underscoring the need for robust clinical trials."
Reply: Dear reviewer, we added the correction in the text.
- Introduction:
- Add more context about the global impact of COVID-19 and why these specific drugs were considered potential treatments.
Reply: Dear reviewer, we added the following excerpt in the text:
The COVID-19 pandemic has had a global impact affecting all spheres of society. In Public Health, it has led to an overload of health systems, loss of human lives, and impact on the mental health of the population [6–8]. In the economic scenario, there was a reduction in consumption and an increase in unemployment. In the social sphere, distancing caused the cancellation of social events. The pandemic has widened social inequalities, exposing the vulnerable population to contagion [6–8]. Due to the potential health catastrophe caused by SARS-CoV-2, a global effort was made to evaluate treatments for COVID-19 to attenuate its global impact, especially by the WHO, with the RECOVERY (Randomized Evaluation of COVID-19 Therapy) international trial [9]. Unfortunately, several countries, such as the United States of America (USA) and Brazil, started to use drugs that showed only in vitro effects against the SARS-CoV-2, such as Ivermectin, Chloroquine/Hydroxychloroquine (CQ/HCQ), and Azithromycin [6,10–12]. Due to the pandemic emergency, the use of well-known substances in order to treat a new disease was a valid option in the beginning, which is called repurposed drugs [13]. Many politicians and healthcare workers who have advocated the early use of these drugs used the argument of lack of time to pursue proper randomized clinical trials (RCTs). Hence, the emergency use of those drugs was justifiable [14]. However, most of the pre-clinical phase I and II studies findings are not confirmed in phase III clinical studies, mainly due to the fact the null hypothesis has a higher probability than the alternative hypothesis being true [14,15].
- Materials and Methods:
Provide more details on the search strategy. For instance, mention the number of studies identified at each stage of the search.
Reply: Dear reviewer, the information is presented in the results sections per drug. The information is indicated in the material and methods section as follows:
In addition, the complete selection of the studies is presented in the result section per drug included in the review.
- Explain the exclusion criteria more clearly, particularly why some studies were excluded. Reply: Dear reviewer, we added the following excerpt in the text:
We excluded the following article types: (i) not related to COVID-19; (ii) not phase III clinical trial; (iii) did not use any of the studied drugs (HCQ/CQ, Ivermectin or Azithromycin) orally; (iv) retracted articles; (v) was not published in English; (vi) did not comprise the date range of the study (COVID-19 pandemic onset to December 2023); (vii) did not evaluate inpatients with COVID-19; (viii) prophylaxis studies; (ix) if we did not have access to the article; (x) if the trial was not registered in online platforms (such as clinicaltrials.com); and (xi) did not present the outcome of interest.
- Results:
- Ivermectin Section:
- Improve clarity by summarizing the results before delving into individual studies. For example, "The majority of studies found no significant benefit of Ivermectin in reducing mortality or severity in COVID-19 patients."
Reply: Dear reviewer, we added the information in all sections.
- In tables, ensure consistency in reporting units (e.g., mg vs. μg/kg).
Reply: Dear reviewer, we added the correction in Tables and text.
- HCQ/CQ Section:
- Provide a similar summary at the beginning of this section. It helps the reader quickly grasp the overall findings.
Reply: We added the summary as suggested.
- Consider a table that compares the outcomes of all three drugs side-by-side for a quick visual reference.
Reply: Dear reviewer, we decided only to include a Graph Summary to have a quick visual reference.
- Discussion:
- Strengthen the discussion by comparing your findings with existing literature, particularly systematic reviews and meta-analyses.
Reply: Dear reviewer, we added minor corrections in the discussion section, and we included the following excerpt:
Due to the absence of an efficacy treatment and the high number of deaths from COVID-19, scientists, physicians, and politicians rush to find a new possible treatment. Perhaps, for physicians, this seeking for a viable treatment might have been even more intense since many of them think “We must give a drug to treat the patient” [14]. This scenario encouraged the adoption of unproven drugs to treat COVID-19, like Ivermectin, CQ/HCQ, and Azithromycin, with no prior RCT, which proves its effectiveness [14]. Many politicians, such as the Brazilian president, and the former American president, publicly endorsed these drugs as silver bullets in the COVID-19 treatment [6,125,126]. Unfortunately, this “infusion of politics into science”, which the international scientific community advised against, occurred worldwide, which might have contributed to the widespread use of these drugs before the publication of RCTs [125–127].
Specifically in Brazil, the sales of the drugs part of the so-called “COVID-Kit”, such as Ivermectin, CQ/HCQ, and Azithromycin, enhanced exponentially. Nearly one in four individuals had taken these drugs [128] which might have contributed to a shortage of these drugs for patients who use them on-label. For instance, in Brazil, HCQ treats rheumatic diseases like lupus or rheumatoid arthritis. In June 2020, pharmacies all over Brazil had a shortage of this drug due to the increased sales to treat COVID-19, which impaired the treatment of rheumatic patients [126]. In the same way, a report in Brazil showed that several individuals who had a prior COVID-19 diagnosis also made use of drugs from the “COVID-19-Kit” being that 19% of them took CQ/HCQ, 55% took Ivermectin, and most of them, that is 77%, took Azithromycin [129]. Not only do these drugs have no efficacy against COVID-19, but they give a “safety” feeling, in which individuals who take them think they are protected against the SARS-CoV-2 infection, leading them to relax measures that are proven to decrease SARS-CoV-2 viral spread, such as social distancing and use of masks, which might have been observed in the DETECT-CoV-2 Brazilian report, that showed a 50% increase of infection in individuals who made use of the COVID-19-kit as prophylactic measures [126,129].
- Highlight the limitations of the studies reviewed, including potential biases and the quality of evidence.
Reply: Dear reviewer, the information is presented in the limitations section as follows:
There is a lot of confusion regarding what the anticipated benefit is likely to be because the literature used several outcomes, such as mortality, disease progression, need for ICU, and reduction in symptoms. Besides the presence of hospitalization, there are disparities of severity among the participants of the study. Discrepancy among the studies for the use of monotherapy vs polypharmacy. Most of the articles included were heterogeneous toward the “standard of care” treatment, which might make difficult comparisons. We only included three widely used drugs to treat COVID-19; we left some other drugs that might have impacted the pandemic management, such as Vitamin D, Oseltamivir, and Nitazoxanide. We did not perform a meta-analysis to assess the cumulative effect of these drugs. We also did not pre-register our systematic review on PROSPERO (International Prospective Register of Systematic Reviews). We only included phase III clinical trials which treated inpatients, this might have limited our analysis to only hospitalized patients.
- Address the broader implications for clinical practice and policy, especially in light of the mixed evidence regarding these treatments.
Reply: Dear reviewer, we added minor corrections in the discussion section to solve the presented issue.
- Conclusion:
- Simplify and focus on the key takeaways. Reiterate that the evidence does not support the efficacy of these drugs in treating COVID-19, based on the reviewed phase III trials.
- Suggest future research directions, such as focusing on newer treatments or more rigorous trials.
Reply: The conclusion section has been corrected as follows:
COVID-19 was one of the deadliest pandemics in modern human history. Due to the potential health catastrophe caused by SARS-CoV-2, a global effort was made to evaluate treatments for COVID-19 to attenuate its impact on the human species. Unfortunately, several countries prematurely justified the emergency use of drugs that showed only in vitro effects against SARS-CoV-2, with a dearth of evidence supporting efficacy in humans. In this context, the purpose of this systematic review was to evaluate the mechanisms of several drugs proposed to treat COVID-19, including Ivermectin, CQ/HCQ, and Azithromycin, and to describe systematically phase III clinical trials that evaluated the efficacy of these drugs for treating patients with this respiratory disease. As the main finding, although Ivermectin, CQ/HCQ, and Azithromycin might have mechanistic effects against SARS-CoV-2 infection, most phase III clinical trials observed no treatment benefit in patients with COVID-19. The evidence doesn´t support the efficacy of these drugs in the treatment of COVID-19, based on the reviewed phase III trials. In this context, future robust studies with greater scientific rigor are needed to focus on new effective treatments.
- Figures and Tables:
- Ensure all figures and tables are clearly labeled and easy to interpret.
Reply: We included minor corrections in the labels.
- Simplify complex diagrams and use color coding or labeling to clarify mechanisms of action.
Reply: Dear reviewer, we added minor corrections in the figures to clarify mechanisms of action.
Reviewer 2 Report
Comments and Suggestions for Authors
This systematic review examines the efficacy of Ivermectin, Chloroquine, Hydroxychloroquine, and Azithromycin against COVID-19 in phase III clinical trials. While the manuscript covers a very interesting topic, it lacks novel outcomes, as the results confirming these drugs' limited effectiveness against COVID-19 are already well-known.
The overall writing and organisation are well-executed. This manuscript could serve as a valuable chronicle of the emergency repurposing of drugs during the early stages of the pandemic.
Comments
1. Contradictory Statements: There are contradictory statements regarding the timeline of the COVID-19 pandemic.
Line 112: "We included phase III RCTs from 2019 to 2023."
Lines 119-120: "The data search was performed on PubMed-Medline, Cochrane, and SciELO from the COVID-19 pandemic onset to December 2023."
The "COVID-19 pandemic onset" was officially declared on 11 March 2020, which conflicts with the inclusion of studies from 2019. Please clarify the timeline or adjust the inclusion criteria.
2. Table 3, Last Row (NCT04358081, 2020): The format of the last entry is inconsistent with the rest of the table. Other studies list the first author's last name et al., or the study's name, but this entry uses only the ClinicalTrials.gov identifier without mentioning the author, study name, or organisation. Please confirm whether this study was obtained from PubMed or other databases mentioned in lines 107-109. If so, revise for consistency.
Italicisation
1. Line 206: The term in vivo should be italicised.
2. Line 1022: The term in vitro should be italicised.
Author Response
Comments and Suggestions for Authors
This systematic review examines the efficacy of Ivermectin, Chloroquine, Hydroxychloroquine, and Azithromycin against COVID-19 in phase III clinical trials. While the manuscript covers a very interesting topic, it lacks novel outcomes, as the results confirming these drugs' limited effectiveness against COVID-19 are already well-known.
The overall writing and organisation are well-executed. This manuscript could serve as a valuable chronicle of the emergency repurposing of drugs during the early stages of the pandemic.
Reply: Dear reviewer, thank you for providing the important comment. In addition, we added the corrections as requested.
Comments
1. Contradictory Statements: There are contradictory statements regarding the timeline of the COVID-19 pandemic.
Line 112: "We included phase III RCTs from 2019 to 2023."
Lines 119-120: "The data search was performed on PubMed-Medline, Cochrane, and SciELO from the COVID-19 pandemic onset to December 2023."
The "COVID-19 pandemic onset" was officially declared on 11 March 2020, which conflicts with the inclusion of studies from 2019. Please clarify the timeline or adjust the inclusion criteria.
Reply: Dear reviewer we adjusted the text. We included only studies from the COVID-19 pandemic onset to December 2023.
Table 3, Last Row (NCT04358081, 2020): The format of the last entry is inconsistent with the rest of the table. Other studies list the first author's last name et al., or the study's name, but this entry uses only the ClinicalTrials.gov identifier without mentioning the author, study name, or organisation. Please confirm whether this study was obtained from PubMed or other databases mentioned in lines 107-109. If so, revise for consistency.
Reply: Dear reviewer, the study was obtained from the ClinicalTrials.gov site. We added the information in the text.
Italicisation
- Line 206: The term in vivoshould be italicised.
Reply: We corrected the term.
- Line 1022: The term in vitroshould be italicised.
Reply: We corrected the term.